# Fragment-sequencing unveils local tissue microenvironments at single-cell resolution

Kristina Handler [1], Karsten Bach[1], Costanza Borrelli [1], Salvatore Piscuoglio[2,3], Xenia Ficht [1], Ilhan E. Acar [1] & Andreas E. Moor [1] ✉

Cells collectively determine biological functions by communicating with each other—both through direct physical contact and secreted factors. Consequently, the local microenvironment of a cell influences its behavior, gene expression, and cellular crosstalk. Disruption of this microenvironment causes reciprocal changes in those features, which can lead to the development and progression of diseases. Hence, assessing the cellular transcriptome while simultaneously capturing the spatial relationships of cells within a tissue provides highly valuable insights into how cells communicate in health and disease. Yet, methods to probe the transcriptome often fail to preserve native spatial relationships, lack single-cell resolution, or are highly limited in throughput, i.e. lack the capacity to assess multiple environments simultaneously. Here, we introduce fragment-sequencing (fragment-seq), a method that enables the characterization of single-cell transcriptomes within multiple spatially distinct tissue microenvironments. We apply fragment-seq to a murine model of the metastatic liver to study liver zonation and the metastatic niche. This analysis reveals zonated genes and ligand-receptor interactions enriched in specific hepatic microenvironments. Finally, we apply fragment-seq to other tissues and species, demonstrating the adaptability of our method.

Biological tissues are multicellular communities arranged in spatially distinct patterns that facilitate efficient cell–cell interaction and maintain tissue homeostasis. Tissue composition can be altered in diseases like cancer where cells interact in different ways to either promote or counteract the diseased state. Single-cell RNA-sequencing (scRNA-seq) has been widely used to characterize cellular compositions in various tissues and diseases. However, during tissue dissociation, a cell's spatial position is lost. To overcome this limitation, recent years have seen rapid technological developments to capture both transcriptomic features as well as corresponding tissue coordinates. Current strategies include imaging-based approaches like in situ hybridization (ISH)[1–3] and in situ sequencing (ISS)[4,5], mRNA capture-based approaches employing arrays of spatial barcodes[6–9], or labeling of cells from distinct spatial areas prior to tissue dissociation[10–12]. However, each of these methods has inherent limitations, for example, unsupervised mRNA capture-based methods usually preclude single-cell analysis, while imaging-based techniques require prior knowledge for targeted panel assembly[13]. In addition, many methods have significant hurdles to their implementation, such as the need for advanced microscopic devices[13,14]. Alternatively, spatially coordinated gene expression has been studied through methods that partially retain native tissue microenvironments such as Paired-cell sequencing[15], PIC-seq[16], and Clump-seq[17], which analyze spatial communities of 2–10 cells together in bulk. Of note, all three of these methods rely on computational deconvolution to approximate single-cell transcriptomes, which is inherently imprecise, especially for genes

[1]Department of Biosystems Science and Engineering, ETH Zürich, Schanzenstrasse 44, 4056 Basel, Switzerland. [2]Institute of Medical Genetics and Pathology, University Hospital Basel, Basel, Switzerland. [3]Visceral Surgery and Precision Medicine Research Laboratory, Department of Biomedicine, University of Basel, Basel, Switzerland. ✉e-mail: andreas.moor@bsse.ethz.ch

that are expressed in multiple cells. Paired-cell sequencing was applied to reconstruct the spatial gene expression of liver endothelial cells (LECs), by selectively enriching for hepatocyte–endothelial cell pairs and exploiting previously established zonated genes in hepatocytes as landmarks[15]. This approach is made possible by the repetitive micro-anatomy of the liver, organized into hexagonal hepatic lobules, in which blood flows from the portal to central veins through sinusoids flanked by linear cords of hepatocytes. Differential oxygen and nutrient availability along the central-portal axis imprint zonation of cell functions and gene expression patterns in liver resident cells, particularly in hepatocytes and LECs[15,18–20]. Furthermore, multiple studies have reported zonation of immune cells, including liver-resident

macrophages called Kupffer cells (KCs), T, and NKT cells, which were all found to be periportally enriched[21,22]. The liver is not only a vitally important metabolic organ but is also frequently affected by metastatic disease, particularly colorectal cancers[23]. However, the impact of metastatic seeding on gene zonation in parenchymal, endothelial, or immune cell subsets remains unclear. In addition, a metastasis-bearing liver may host additional infiltrating adaptive and innate immune cells, such as effector T cells, monocytes, and macrophages, which shape the tumor microenvironment (TME)[24]. Characterizing the composition of the TME, as well as the changes in cell and gene zonation in the metastases-bearing liver are therefore interesting research questions for spatial transcriptomics.

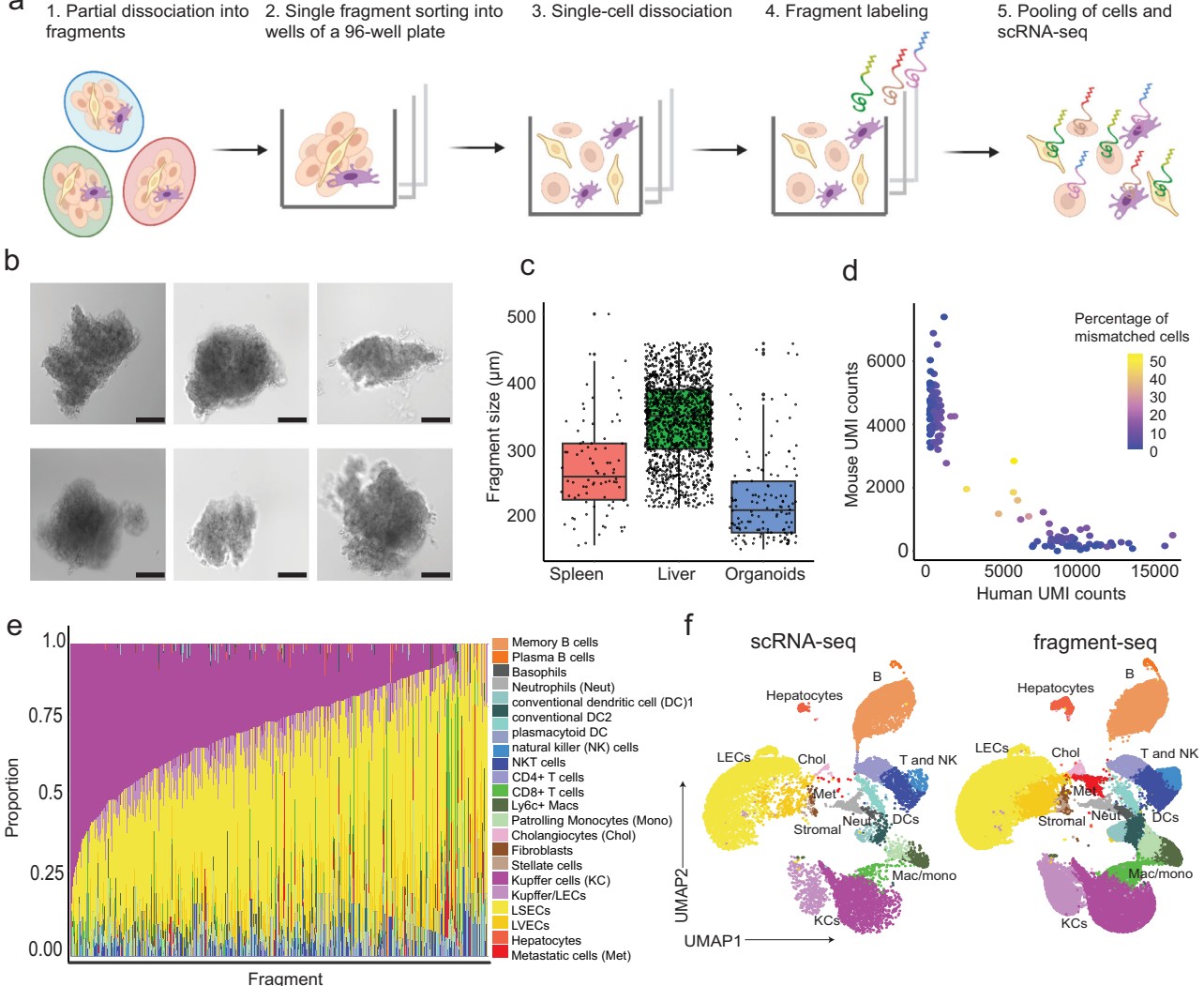

**Fig. 1 | Overview of fragment-seq and assessment of quality and accuracy.**
**a** Schematic illustration of fragment-seq workflow. scRNA-seq: single-cell RNA-sequencing. Created with BioRender.com. **b** Brightfield microscopy images of representative liver fragments. Scale bar: 100 μm. Imaging was performed for 9 96-well plates in one experiment with results shown in Supplementary Fig. 1b, here six representative fragments are shown. **c** Fragment-size distribution of sorted fragments from different tissues visualized in boxplots [$n = 82$ (murine spleen), 1468 (murine liver), 138 (CRC organoid) fragments]. Black dots indicate fragments. For box and whiskers plots the middle line represents the median; the upper and lower lines are the first and third quartile (Q1 and Q3); the whiskers indicate the upper and lower limits of data spread by subtracting 1.5* interquartile range (IQR) from Q1 and adding 1.5* IQR to Q3. **d** Scatter plot showing the fraction of mismatched cells. Dots represent fragments and their color indicates the percentage of mismatched cells

(human cells within mouse fragments and mouse cells within human fragments) ($n = 139$ fragments from 1 sample). UMI unique molecular identifier. **e** Barplot showing cell type proportions per fragment; only fragments with at least 5 cells are considered ($n = 1568$ fragments from a total of 10 samples). LEC liver endothelial cell, LSECs liver sinusoidal endothelial cells, LVECs lymphatic vascular endothelial cells. MAC macrophages **f** Uniform manifold approximation and projection (UMAP) visualization of batch-corrected (see the "Methods" section), single-cell transcriptomes from fragment-seq and scRNA-seq of mouse metastatic liver samples. Cells are clustered, annotated, and colored by their cell type. Cells are separated based on the protocol [conventional scRNA-seq ($n = 2$ samples) and fragment-seq ($n = 10$ samples)]. For **c** and **d** the source data are provided as a Source Data file.

Here, we introduce fragment-sequencing (fragment-seq), enabling the transcriptomic characterization of single cells within spatially distinct tissue niches, which are partitioned using a large fragment biosorter and later reconstructed based on landmark genes or cell types. Fragment-seq was inspired by previously published approaches, such as Paired-cell sequencing[15], PIC-seq[16], and Clump-seq[17]. It refines these approaches and achieves single-cell resolution while simultaneously capturing larger communities of cells than its predecessors, thus reflecting biologically relevant tissue microenvironments in three-dimensional space. For greater analytical power, fragments are grouped based on landmark genes or the presence/absence of one cell type to represent distinct spatial niches. Thus, fragment-seq exploits the spatial proximity of landmark cells or genes to other cell types to more accurately reconstruct tissue niches, without assigning a specific physical location to a cell or transcript. Importantly, this enabled us to predict ligand–receptor (L–R) interactions which are not only significantly enriched in a scRNA-seq dataset, but actually co-occur in specific microenvironments. To this end, fragment-seq combines previously established methods of single object sorting using a large fragment biosorter, cell hashing using lipid-tagged barcodes[25], and scRNA-seq[26,27]. We apply fragment-seq to dissect spatially distinct niches in the metastasis-bearing mouse liver and shed light on the importance of immune zonation by reporting zonated gene expression in KCs in the metastasis-carrying liver. Furthermore, we identify spatially restricted L–R interactions involving different macrophage subsets within hepatic and metastatic compartments. These results highlight the potential of fragment-seq to capture spatial heterogeneity within complex tissues.

## Results

### Fragment sorting with a large fragment biosorter enables the characterization of individual cellular communities

Fragment-seq is based on partial tissue dissociation followed by size-dependent sorting of individual Ø 200–450 μm cellular communities (fragments) with a large fragment biosorter into single wells of a 96-well plate; one fragment per well. Subsequently, fragments are dissociated into single cells within each well and hashed with a fragment-specific lipid-tagging barcode[25] to preserve information about a cell's neighborhood prior to pooling for scRNA-seq (Fig. 1a). The accuracy of the sorting approach was assessed by sorting, plate imaging, and counting of fragments per well. The majority of wells contained a single fragment, and multiple fragments per well were rare, therefore we concluded that the sorting process is sufficiently accurate (Supplementary Fig. 1a, b). Due to partial dissociation, fragments may vary in shape (Fig. 1b). However, fragments can be size-gated during sorting by estimating the size from a regression of the acquired time-of-flight (TOF) on the size of standardized beads (Fig. 1c, Supplementary Fig. 1c). To assess whether cells assigned to the same fragment truly derive from the same cellular neighborhood, we performed a species-mixing experiment: we mixed GFP+ mouse and GFP− human colorectal cancer (CRC) organoids to represent fragments. Using fluorescent index sorting we then sorted 144 wells with GFP+ and 144 wells with GFP− organoids. We then applied the fragment-seq procedure, mouse or human unique molecular identifiers (UMIs) were used to identify the species of origin for each cell, and the percentage of cells that reflected the expected or mismatched species based on GFP-signal from indexed sorting was plotted (Fig. 1d, Supplementary Fig. 2a, b). This revealed that 95% of cells were accurately assigned (Supplementary Fig. 2c). Altogether this confirmed the validity of the fragment-seq approach. Next, we applied both fragment-seq and conventional scRNA-seq to mouse liver samples harvested 2 weeks after intrasplenic injection of murine CRC organoids. Resulting single-cell transcriptomes were annotated using known marker genes from the liver cell atlas[22], assigned to their fragment of origin, and different cell type proportions per fragment were plotted (Fig. 1e, Supplementary

Fig. 3a–c). Comparison of fragment-seq data with conventional scRNA-seq data of murine metastatic livers showed a similar distribution of cell types and cellular quality (Fig. 1f, Supplementary Fig. 3d).

### Reconstruction of fragment position along the central–portal axis based on zonated gene expression

To reconstruct the native position of a fragment along the central–portal axis (lobule layers L1–L10) we used a previously identified LEC zonation-specific gene expression signature[15]. For each fragment, a zonation coordinate between 0 (=central) and 1 (=portal) was calculated based on the average expression of periportal landmark genes (pLM) in LECs divided by the sum of the average expression of periportal and pericentral landmark genes (cLM) (Fig. 2a). Based on zonation coordinates, fragments were assigned to a lobule layer or simply grouped into pericentral and periportal zones (Supplementary Fig. 4a). The accuracy of the reconstruction was corroborated by zonated gene expression in hepatocytes from pericentral and periportal fragments which matched previously published landmark genes[20] (Supplementary Fig. 4b). This spatial ordering allowed us to characterize differentially expressed genes in LEC from different zones, including genes that were not reported as zonated in the reference dataset (For example for *Plpp1*, mean expression across lobule layers: L1–3: 2.64, L4: 2.63, L5: 2.13, L6: 1.39, L7: 0.991, L8-10: 0.739, with respective standard deviation [SD] L1–3: 1.65, L4: 1.34, L5: 1.23, L6: 1.06, L7: 0.948, L8-10: 1.16, was found to be significantly zonated using a two-sided empirical Bayes quasi-likelihood *F*-test, *p*-value = 1.18e−18; as was *Galnt15*, mean expression across lobule layers: L1–3: 0.81, L4: 0.637, L5: 0.987, L6: 1.78, L7: 3.14, L8-10: 4.51 with an SD of L1–3: 0.538, L4: 1.24, L5: 1.31, L6: 1.58, L7: 2.35, L8-10: 3.79, *p*-value = 2.17e-37) (Fig. 2b, c). Some of the identified genes were also zonated in LEC from healthy mouse liver and are therefore likely involved in homeostatic processes (Supplementary Fig. 4c). For example *Plpp1*, a phospholipid phosphatase, was centrally zonated (mean expression portal: 1.50, central: 2.39, SD portal: 0.964, central: 1.21, two-sided Wilcoxon signed-rank test, *p*-value = 5.5e−07)−implying a link to pericentral lipogenesis[18]−while *Galnt15*, putatively involved in O-linked oligosaccharide biosynthesis, was portally zonated (mean expression portal: 0.573 and central: 0.224 with SD portal: 0.520, central: 0.280, two-sided Wilcoxon signed-rank test, *p*-value = 4.4e−06), which is in accordance with periportal gluconeogenesis[18]. Next, we performed molecular cartography (MC), a highly-multiplexed fluorescence in-situ hybridization (FISH) approach, with a custom 100-gene panel that included marker genes for key cell types, as well as genes that displayed a spatially variable expression pattern in fragment-seq (Supplementary Fig. 5a–d). This validated zonation patterns of metabolic genes in LECs (*Plpp1* mean expression portal: 58.2 and central: 134, with SD portal: 30.9 and central: 43.4, was found to be significantly zonated using a two-sided empirical Bayes quasi-likelihood *F*-test, *p*-value = 4.30e−49, as was *Galnt15*, with mean expression portal: 99.2 and central: 31.5, SD portal: 81.6 and central: 38.2, *p*-value = 9.94e−26) (Fig. 2b,d; Supplementary Fig. 4d). Moreover, we could validate zonated gene expression of *Plpp1* and *Galnt15* in publicly available Visium datasets from healthy and non-alcoholic fatty liver disease (NAFLD) murine livers[22] (Supplementary Fig. 4g, h). Fragment-seq furthermore permitted the investigation of zonated genes in other cell types after assigning a fragment to periportal or pericentral zones according to LEC gene expression patterns. For example, *Vcam1* was found to be significantly enriched in periportal KCs using a two-sided empirical Bayes quasi-likelihood *F*-test (*p*-value = 2.13e−06, mean expression across lobule layers L1–3:4.578, L4: 5.62, L5: 6.13, L6: 8.56, L7: 11.415, L8-10: 10.36, with respective SDs of L1–3: 4.258, L4: 3.917, L5: 3.89, L6: 5.498, L7: 6.939, L8-10: 6.454) (Fig. 2e). This could be independently validated by MC (two-sided empirical Bayes quasi-likelihood *F*-test, *p*-value = 4.44e−05, average portal expression of 107 with a SD of 77.6, while average central

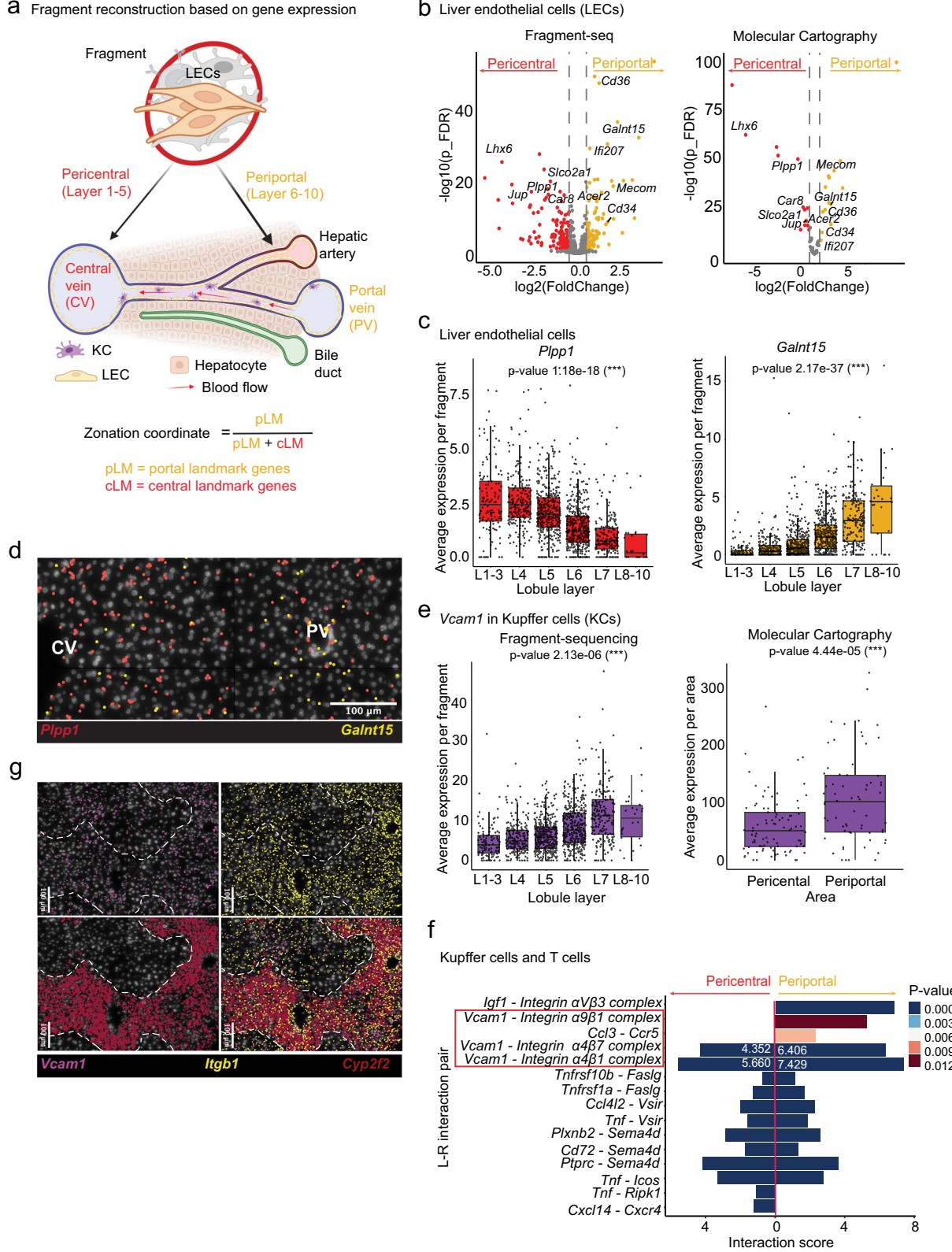

**a** Fragment reconstruction based on gene expression

**b** Liver endothelial cells (LECs)

**c** Liver endothelial cells

**d**

**e** *Vcam1* in Kupffer cells (KCs)

**g**

**f** Kupffer cells and T cells

expression was 59.1, SD = 48.7) (Fig. 2e). Interestingly, we did not find *Vcam1* to be significantly zonated in KCs in healthy livers (mean portal: 99.2, central: 31.5 and SD portal: 81.6, central: 38.2, not significant [NS] in a two-sided non-parametric Wilcoxon signed-rank test), suggesting its upregulation is a response to an inflammatory process (Supplementary Fig. 4e). This is in line with *Vcam1* being more heavily expressed and zonated in NAFLD compared to healthy livers in

previously published data[22] (Supplementary Fig. 4g, h). However, due to the lack of single-cell resolution of Visium data, it cannot be directly determined whether zonated gene expression is caused by KCs or LECs. In fact, prior knowledge of the importance of *Vcam1* in LECs would likely lead to neglecting its zonation in KCs. Upregulation of VCAM-1 during metastatic disease has been previously reported for LECs and may contribute to immune cell recruitment by interactions

**Fig. 2 | Fragment-seq application to investigate gene zonation during liver metastasis. a** Schematics of liver microanatomy and fragment zonation approach. Created with BioRender.com. **b** Differentially expressed genes (DEGs) of LECs in spatially ordered fragments. Left, fragment-seq (1384 fragments across 9 samples). Right, Molecular Cartography (MC) (155 areas across 4 samples). Colored dots represent significantly enriched genes; red, enriched in pericentral zones; yellow, enriched in periportal zones. Gene labels indicate genes significantly enriched in both analyses. p_FDR: false discovery rate adjusted *p*-value. **c** Boxplots showing gene expression in LECs of spatially ordered fragments (*n* = L1–L3: 137, L4: 214, L5: 402, L6: 409, L7: 196, L8–L10: 26 fragments across 9 samples). Black dots represent individual fragments. **d** Representative images of MC of *Plpp1* (red) and *Galnt15* (yellow) are shown as an overlay with DAPI signal (white). CV central vein, PV portal vein. **e** Boxplots of *Vcam1* gene expression in Kupffer cells (KCs) of spatially ordered fragments or spatial areas. Left, fragment-seq (like in c) (*n* = L1–L3: 115, L4: 206, L5: 379, L6: 387, L7: 182, L8–L10: 26 fragments across 9 samples); right, MC comparing pericentral and periportal. **f** Predicted ligand–receptor (L–R) interactions between KCs and T cells in grouped fragments from pericentral or periportal

origin (*n* = 9 samples). Interaction scores were calculated from grouped fragment-seq data by CellPhoneDB, which uses a permutation test to generate *p*-values (unadjusted) indicating significantly enriched L–R interactions. Interactions referenced in the main text are highlighted with red squares and white numbers indicate interaction scores. **g** Representative MC of *Vcam1* (purple), *Itgb1* (yellow), and *Cyp2f2* (red) shown as an overlay with DAPI signal (white). For MC in **d**, **e**, and **g** the complete dataset (as shown in d) was from pericentral *n* = 89; periportal *n* = 66 areas across four samples from four separate experiments. For **b**, **c**, and **e**, we used a negative binomial generalized log-linear model ('glmQLFTest' function of edgeR), which uses a (two-sided) empirical Bayes quasi-likelihood *F*-test. *P*-values (Benjamini–Hochberg adjusted) of <0.05 were considered significant (***<0.001). For **b**, **c**, **e**, and **f** the source data are provided as a Source Data file. For all box and whiskers plots the middle line represents the median; the upper and lower lines are the first and third quartile (Q1 and Q3); the whiskers indicate the upper and lower limits of data spread by subtracting 1.5* interquartile range (IQR) from Q1 and adding 1.5* IQR to Q3. L–R: ligand–receptor.

with its binding partner integrin α4β1[28]. To date, the function of VCAM-1 in KCs has not been well studied, but Okada et al. have described that VCAM-1 can mediate interactions of KCs and lymphocytes, which in turn promotes KC activation[29]. This spurred us to perform L–R interaction analysis[30] for KCs and lymphocytes (T and B cells) in the periportal and pericentral zones. This analysis predicted an enrichment of interactions of the KC-expressed ligand VCAM-1 with α9β1, α4β7, and α4β1 integrin complex receptors on T and B cells in periportal compared to pericentral zones (Fig. 2f, Supplementary Fig. 4f). MC could validate an enriched periportal gene expression of *Vcam1* and *Itgb1* (Integrin subunit β1) (Fig. 2g). A primary function of α4β7 and α4β1 integrins on T cells is mediating tissue homing and adhesion to the vessel wall[31], indicating that periportal KCs may have upregulated *Vcam1* expression to promote lymphocyte recruitment. This is in line with an increase in predicted *CCL3|CCR5* interactions between periportal KCs and T cells, as CCR5 was reported to mediate the recruitment of T cells to the liver during acute inflammation[31]. Additionally, we could confirm the upregulation and zonation of *Vcam1|Itgb1* and *Ccl3|Ccr5* in NAFLD livers, suggesting that upregulation of these ligands and receptors is a response to inflammatory liver perturbations in general (Supplementary Fig. 4g, h). Next, we set out to assess potential biases caused by different fragment sizes and fragment cell counts. First, we compared pericentral and periportal fragments and found them to be comparable in size (Supplementary Fig. 6a), while cell counts were on average slightly lower in periportal than in pericentral fragments (Supplementary Fig. 6b). In order to test for biases introduced by fragment size or cell number, we divided the dataset according to fragment size into small (211–325 μm) or big fragments (326–457 μm) or applied different minimum cutoffs for cell numbers (≥5 or ≥20 cells per fragment). Next, we performed differential gene expression (DGE) analysis for LECs or KCs, which did not reveal any significant changes (Supplementary Fig. 6c,d). A similar pattern of significantly upregulated zonated genes could be found in all scenarios (Supplementary Fig. 6e). Moreover, recovered cell types and predicted L–R interactions were similar between different cutoffs, with only a few L–R interactions not being replicated across all size gates (Supplementary Fig. 6f and Supplementary Fig. 7a). In order to assess whether this effect is caused by differential fragment sizes, or is rather a consequence of undersampling, we compared L–R interactions from the complete pericentral and periportal datasets with datasets from small fragment sizes (211–325 μm), and a downsampled dataset, representing the same amount of cells as the small fragment size data, but covering the complete size range (211–457 μm) (Supplementary Fig. 7b). This revealed that interactions such as *Vcam1|Itga9* and *Cxcl14|Cxcr4* were lost in both analyses (211–325 and 211–457 μm downsampled), indicating that the overall number of acquired fragments is more critical to the robustness of results than the consistency of

fragment sizes. Therefore, we recommend prioritizing sample size over restrictive gating or subsampling of fragment sizes. All in all, this demonstrated that neither fragment size nor the number of cells recovered per fragment significantly impacted our findings, at least not within the given cutoffs of ≥5 cells per fragment and 200–450 μm diameter.

## Fragments from metastatic-proximal and -distal sites reveal differences in local microenvironments

Instead of grouping fragments according to liver zonation, we next grouped them based on their relation to metastatic sites. Fragments containing metastatic cells were defined as 'proximal' and fragments without metastatic cells as 'distal' to metastatic sites (Fig. 3a, b). Of note, we only included samples from mice with visible metastasis and a sufficient number of recovered metastatic cells in this analysis, to exclude animals with very low tumor burden (Supplementary Fig. 8a, b). Groups did not show any significant variability in fragment size or cell numbers per fragment (using a two-sided Wilcoxon signed-rank test, mean size distal: 355.1 μm and proximal: 349.0 μm with SD distal: 58.8 μm and proximal: 58.9 μm, *p*-value = 0.3, and for cell counts we obtained means of distal: 28.3 cells and proximal: 31.2 cells, with SD distal: 17.5 cells and proximal: 20.1 cells, *p*-value = 0.46) (Supplementary Fig. 8c, d). We then assessed cell-type proportions of key cell subsets and found that proximal areas had a significantly higher proportion of macrophages/monocytes (mean proportion of 0.04 ± 0.01 distal and 0.2 ± 0.01 proximal, *p*-value = 6.17e−08, two-sided empirical Bayes quasi-likelihood *F*-test) and metastatic cells (mean proportion of 0.00 ± 0.00 distal and 0.09 ± 0.01 proximal, *p*-value = 2.62e−10, two-sided empirical Bayes quasi-likelihood *F*-test), while KCs (mean proportion of 0.40 ± 0.05 distal and 0.27 ± 0.02 proximal, *p*-value = 4.72e−06, two-sided empirical Bayes quasi-likelihood *F*-test) and LECs (mean proportion of 0.43 ± 0.05 distal and 0.30 ± 0.02 proximal, *p*-value = 6.04e−06, two-sided empirical Bayes quasi-likelihood *F*-test) were reduced (Fig. 3c, Supplementary Fig. 8e). Segmentation and cell-type annotation of MC data validated those findings (Fig. 3d, e; Supplementary Fig. 8e), highlighting that fragment-seq faithfully recapitulates in situ cell-type proportions. We decided to investigate the macrophage/monocyte subset more closely and found that proximal areas showed a trend for the enrichment of *C1q*+ macrophages (Fig. 3f, Supplementary Fig. 3c). We could validate the increased expression of complement genes (*C1qb*, *C1qc*) in macrophages/monocytes within proximal areas in MC data (Fig. 3g). *C1q*+ macrophages are reportedly involved in T cell exhaustion and are an indicator of poor prognosis in many cancers[32]. Therefore, we wanted to investigate cellular crosstalk between macrophages/monocytes and T cells further. Due to constraints in cell numbers of fragment-seq data and cell type markers included in MC, we did not further differentiate between T or

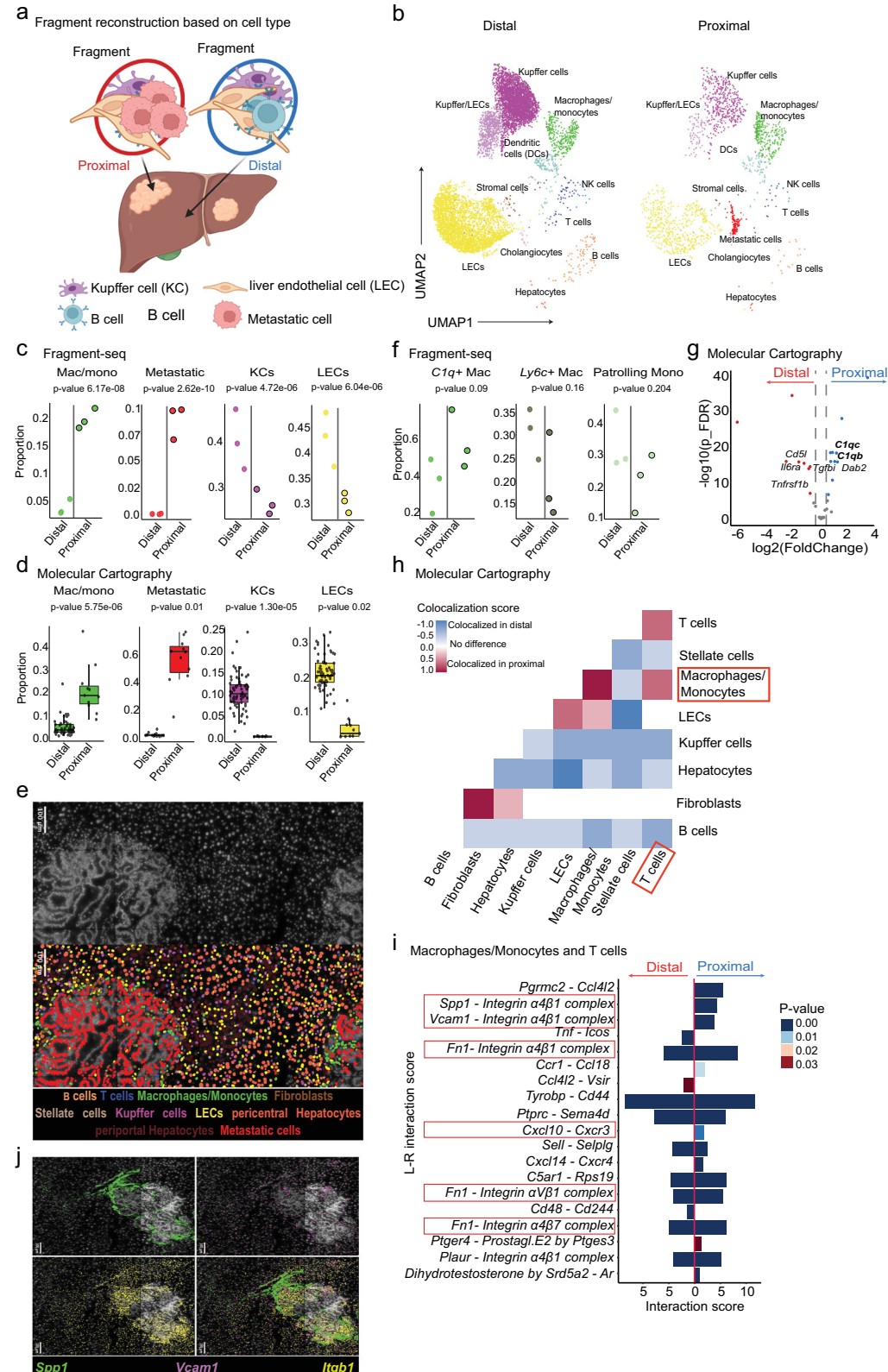

macrophage cell subsets. We found significantly more colocalization of macrophages/monocytes and T cells within proximal areas in MC data (Fig. 3h, Supplementary Fig. 8f) suggesting a potential role for macrophage/monocyte|T cell interactions in shaping the immune response in the metastatic microenvironment. To identify potential drivers of these interactions we performed L−R interaction analysis[30] on fragment-seq data which revealed a number of spatially enriched

interactions (Fig. 3i). Notably, this included interactions of Secreted Phosphoprotein 1 (*Spp1*), Fibronectin-1 (*Fn1*) and *Vcam1* on macrophages/monocytes with α4β1 integrin complexes on T cells. Their proximal enrichment could be validated by MC (Fig. 3j). Additionally, we could validate the proximal enrichment of SPP1 and VCAM-1 protein expression with immuno-fluorescence (IF) (Supplementary Fig. 8g). While VCAM-1|α4β1 interactions are mostly considered to be

**Fig. 3 | Fragment-seq application to investigate local differences in metastatic-proximal and -distal microenvironments. a** Schematics of reconstructing fragment position based on the presence of metastatic cells (distal = fragments without metastatic cells; proximal = fragments with metastatic cells). Created with BioRender.com. **b** Separated UMAPs based on reconstructed groups from integrated samples ($n = 3$). Cells are clustered, annotated, and colored by their cell type. **c** Dot plots representing cell type proportions of grouped fragment-seq data ($n = 3$ samples). From left to right: macrophages/monocytes (Mac/Mono), metastatic cells, KCs, and LECs. Dots represent individual mice and dots with black circles represent grouped proximal fragments. **d** As in **c**, but dots represent areas from molecular cartography (MC). **e** Representative MC images; upper image: DAPI (white); lower image, DAPI stain overlaid with cell type annotations. **f** As in **c**, from indicated Mac/Mono subtypes ($n = 3$ samples). **g** Differentially expressed genes (DEGs) of macrophages/monocytes between distal and proximal regions from MC highlighted in a volcano plot. Colored dots represent significantly enriched genes; blue, enriched in proximal; red, enriched in distal. p_FDR: false discovery rate adjusted *p*-value. **h** Cell colocalization map built from MC data comparing the

frequency of colocalization in distal and proximal areas using a two-sided permutation test, no correction for multiple comparisons. **i** Predicted ligand–receptor (L–R) interactions between macrophages/monocytes and T cells in distal or proximal areas based on fragment-seq data ($n = 3$ samples). Interaction scores were calculated from grouped fragment-seq data by CellPhoneDB using a permutation test (unadjusted *p*-value indicated). **j** Representative MC image of *Spp1* (green), *Vcam1* (purple), and *Itgb1* (yellow) shown as an overlay with DAPI signal (white). MC data in **d**, **e**, **g**, **h**, and **j** represents two samples from two independent experiments from mice with visible macrometastases ($n$ = distal: 71, proximal: 11 areas). For **c**, **d**, **f**, and **g**, we used a negative binomial generalized log-linear model ('glmQLFTest' function of edgeR), which uses a (two-sided) empirical Bayes quasi-likelihood *F*-test. *P*-values (Benjamini–Hochberg adjusted) of <0.05 were considered significant. For, **c**, **d**, and **f**–**i** the source data are provided as a Source Data file. For all box and whiskers plots the middle line represents the median; the upper and lower lines are the first and third quartile (Q1 and Q3); the whiskers indicate the upper and lower limits of data spread by subtracting 1.5* interquartile range (IQR) from Q1 and adding 1.5* IQR to Q3. L–R: ligand–receptor.

adhesive interactions, SPP1 and FN1 are associated with a poor prognosis in various cancer types[33–35]. SPP1 was previously found to suppress T cell responses in the TME[36], while *Fn1* reportedly correlates with the infiltration of anti-inflammatory macrophages[35]. Interactions of both of these macrophage/monocyte-derived ligands with integrins on other immune cells were found to be enriched in colorectal cancer and might be involved in promoting tumorigenesis[37]. In addition, we found a proximal enrichment of *Cxcl10|Cxcr3*, an interaction that is well-known for mediating effector T cell recruitment[38] and has been found to be enriched between *C1q+* macrophages and other immune cell subsets in colorectal cancer[37].

**Application of fragment-seq to other tissues and comparison to Visium and MC**

Fragment-seq can be easily adapted to other tissue types and species with minor adjustments to the single-cell dissociation protocol. We have created proof-of-principle datasets from the mouse spleen (Supplementary Fig. 9a–c) and further showed that the method is compatible with clinical samples by applying it to Crohn's disease biopsies (Supplementary Fig. 9d–f). For clinical samples, we introduced size selection based on sequential filtering and manual picking of fragments to demonstrate the feasibility of the protocol in the absence of an expensive biosorter apparatus (Supplementary Fig. 9g).

Direct comparison to Visium[39], one of the most widely used spatial transcriptomics methods, demonstrates the strength of fragment-seq. Visium's in situ permeabilization protocols need to be optimized for different tissue compositions, resulting in compromised solutions and uneven data quality in heterogeneous tissues such as metastasis-bearing organs, while fragment-seq demonstrates far more consistent results (Supplementary Fig. 10a-c). The greatest advantage of fragment-seq compared to slice-based spatial transcriptomics is single-cell resolution. In contrast, Visium data requires deconvolution of spots into single cells, which revealed a clear bias towards hepatocytes in our datasets (Supplementary Fig. 10d). This phenomenon has been previously described[40] and is attributed to the RNA content of hepatocytes superseding that of smaller cells like LECs. As described before, we could show zonated gene expression in publicly available Visium datasets (Supplementary Fig. 4g, h), however, the lack of single-cell resolution precludes reliable allocation of gene expression to one specific cell type. In sum, comparison with Visium demonstrates fragment-seq's superior performance in uncovering zonated gene-expression patterns in less frequent cell types. Comparison of fragment-seq to the imaging-based spatial transcriptomics approach MC revealed that fragment-seq relatively overrepresented KC, and LECs, while underrepresenting cell types like hepatocytes, and stromal or metastatic cells (Supplementary Fig. 10e).

**Combining slice-based spatial transcriptomics and fragment-seq for refined reconstruction of spatial niches**

Like any methodology that uses spatial reconstruction, fragment-seq mainly relies on prior knowledge of landmark gene expression patterns[41] to determine the spatial origin of fragments within tissues. As proof of principle, we explored the possibility of using landmark genes derived from MC datasets to sort fragments into their niches of origin. To this end, we filtered MC datasets from distal and proximal areas of metastases-bearing mice for cell types that are readily represented in fragment-seq (i.e. B cells, KCs, LECs, metastatic cells, macrophages/monocytes, and T cells). Next, we identified the top ten differentially expressed genes in distal and proximal areas, resulting in a set of 20 landmark genes (Supplementary Fig. 11a). Subsequently, we transformed fragments from a single mouse (to mitigate batch effects) into pseudobulks, clustering them exclusively using the 20 established landmark genes, which unveiled two distinct clusters of fragments (Supplementary Fig. 11b). Feature plots confirmed the differential expression of top landmark genes, with distal landmark genes predominantly enriched in cluster 1 and proximal landmark genes in cluster 2 (Supplementary Fig. 11c). Finally, we assessed the cell type proportions among fragments from each cluster, which confirmed that cluster 2 was enriched in metastatic cells (Supplementary Fig. 11d). We envision that using more agnostic and comprehensive slice-based spatial transcriptomics methods (for example laser capture microdissection or array-based approaches) could be used to build even more refined reference maps for spatial landmarks in different tissues and diseases.

## Discussion

Here, we present fragment-seq, a method that provides single-cell transcriptomes from cellular communities while allowing the reconstruction of their anatomical niche of origin based on the abundance of landmark genes or cell types. By applying fragment-seq to the perturbed mouse liver we showed that we can capture different microenvironments, i.e. lobule layers along the central-portal axis or areas distal and proximal to metastatic sites. For example, this revealed that periportal KCs upregulate *Vcam1* expression and L–R interaction analysis predicted an increased involvement of VCAM-1 ligand interacting with integrin receptors on T and B cells periportally. We hypothesize that increased *Vcam1* expression could serve to boost immune cell recruitment from the portal blood in order to counteract the perturbation after injection of metastatic cells[42].

To demonstrate that fragments can be assigned to a microenvironment of origin by using landmark cell types, we grouped fragments into proximal or distal to metastatic sites according to the presence or absence of metastatic cells. This demonstrated that

fragment-seq is a powerful tool to probe the metastatic microenvironment. To confirm the correct assignment of fragments to specific niches, future experiments would require anchoring labels that are directly linked to the spatial location within the tissue of origin. Unlike other methods that allow the investigation of spatial metastatic niches[11,43], fragment-seq does not rely on transgenic mice to induce a fluorescent reporter or tissue slicing and photoactivation of regions of interest. Fragment-seq allows the investigation of different metastatic niches in a three-dimensional, high throughput, and unsupervised way. For example, we uncovered an increased abundance of *C1q*-expressing macrophages in metastatic-proximal sites, which were previously described as a subset that is increased in cancer correlating with poor prognosis and T cell exhaustion[32]. L–R interaction analysis further allowed us to identify proximal enrichment of interactions involving *Spp1* or *Fn1*, which were previously found to be associated with CRC tumor progression[37]. These biological findings highlight that fragment-seq can be used to capture small local spatial differences within complex tissues, while its single-cell resolution allows niche-specific L–R interaction analysis.

We could further demonstrate in two proof-of-principle experiments that fragment-seq can be easily adapted to other biological tissues and species, as shown for the murine spleen and human Crohn's disease biopsies. For the latter, we established a manual sorting approach, to make fragment-seq feasible also in the absence of a large fragment biosorter. Our preliminary data show that fragment-seq can be applied to fresh human tissues, which could be potentially used to address fundamental questions about disease mechanisms in humans. For example, fragment-seq could distinguish fibrotic and non-fibrotic microenvironments in Crohn's disease samples based on the presence or absence of activated fibroblasts or could be used to separate pro- or anti-inflammatory microenvironments within solid tumors. Of note, fragment-seq is limited by cell loss, therefore, small tissue samples (~<1 cm$^3$) may require pooling.

However, like any spatial transcriptomics methodology, fragment-seq has inherent strengths and limitations. Single-cell resolution is one of its strongest advantages because it enables precise L–R interaction prediction within distinct spatial locations. Another advantage of fragment-seq is its compatibility with existing downstream protocols used for scRNA-seq, such as the vast existing catalog of computational methods. Furthermore, fragment-seq has the potential to profile other modalities, such as chromatin accessibility or proteins, solely by adapting the scRNA-seq protocol to ATAC-seq[44,45] or CITE-seq[46] approaches, respectively. So far, fragment-seq has been exclusively applied to fresh tissues, however, the protocol could be modified for frozen tissues. Fragment-seq could theoretically be adapted to single nuclei RNA sequencing by modifying lysis protocols and exchanging lipid- with cholesterol-tagging barcodes[25]. Unlike sliced-based spatial transcriptomics methods, fragment-seq is not limited to individual two-dimensional regions of interest and is, therefore, more representative of the sample tissue as a whole. In addition, fragment-seq is not limited by a panel of marker genes and does not require digital segmentation to achieve single-cell resolution, which is a clear advantage compared to many sliced-based spatial transcriptomics methods such as MC. However, fragments are generated in a random process and rare niches might not be reliably captured. This could be amended by introducing fluorescently labeled antibodies or fluorescent reporter genes to indicate rare niches of interest and select signal-positive fragments with the large sample sorter. Alternatively, it may be possible to combine fragment-seq with proximity-labeling systems such as sLP-mCherry (which is based on cell-permeable mCherry that is secreted by a sender cell to integrate into the cell membrane of neighboring cells[43]) or recently developed cell–cell contact tracing models[47], e.g. to label tumor-proximal areas with fluorescent signals that could be selectively enriched with the biosorter. As with any other spatial or scRNA-seq methods cell type biases are introduced by fragment-seq. Certain cell types like stromal cells and hepatocytes were underrepresented compared to imaging-based MC. However, a comparison of fragment-seq with conventional scRNA-seq suggests that both share similar biases, possibly introduced due to ex vivo liver digestion[22], in this regard, fragment-seq and scRNA-seq data are comparable.

As demonstrated here, fragment-seq enables unsupervised spatial hypothesis generation for entire tissues. This makes fragment-seq an ideal method for exploratory research, which can then be validated by imaging-based spatial transcriptomics or proteomics methods that offer subcellular spatial resolution but require prior knowledge of gene expression for panel assembly. In sum, we show that fragment-seq is a powerful tool to investigate differences in cellular composition and gene expression between distinct tissue niches and serves as a valuable addition to currently available technologies in the field of spatial transcriptomics.

## Methods

### Ethics statement

All experimental procedures were performed in accordance with Swiss Guidelines. Animal experiments were approved by the Cantonal Veterinary Office Basel-City under the national license number 35370. Ethical approval for collecting tissue specimens from IBD patients was received by the Cantonal Ethics Committee of Canton Zürich (BASEC-No. EK-1755/ PB_2019-00169). Ethical approval for collecting tissue specimens from colorectal cancer patients was received by the Basel Ethics Committee (EKBB, approval number 2019-00816). Written informed consent was collected before study inclusion from all participants.

### Mice

For all experiments C57BL/6J mice were obtained from Janvier-Labs (Le Genest-Saint-Isle, France), mice were used aged 6–10 weeks, and male and female mice were used interchangeably. Animals were housed on a 12 h day–night cycle, with ad libitum access to a standard diet and drinking water in a room with controlled temperature (21 ± 2 °C) and humidity (55 ± 10%). For experiments with liver metastases, the Cantonal authorities under this license foresee humane endpoints in case of weight loss of more than 15%, signs of infection, a palpable tumor diameter of more than 1.5 cm, or the presence of severe behavioral anomalies indicating pain. These limits have not been exceeded in this study.

### Mouse model of liver metastasis

VilCreERT2;APCfl/fl;Tp53fl/fl;KrasG12D/wt (AKP) organoids, obtained from Owen Sansom from Beatson Institute for Cancer Research in Glasgow, were modified with an additional knockout in Smad4, resulting in VilCreERT2;APCfl/fl;Tp53fl/fl;KrasG12D/wt;Smad4KO (AKPS). Organoids were cultured in Matrigel (Corning) following standard protocols in the field[48,49]. The culture medium (Advanced DMEM/F12, Life Technologies™) was supplemented with 10 mM HEPES (Life Technologies™), 2 mM L-Glutamine (Life Technologies™), 100 mg/mL Penicillin/streptomycin, 1x B27 supplement (Life Technologies™), 1x N2 supplement (Life Technologies™) and 1 mM N-acetylcysteine (Sigma-Aldrich). Organoids were split by mechanical dissociation every 3 days. For splenic injections, AKPS organoids (4 domes per mouse) were repeatedly washed in ice-cold PBS to remove all Matrigel, then mechanically dissociated into small fragments, resuspended in 50 µL PBS, and loaded in an insulin syringe (BD, MicroFine, 0.3 ml, 30 G). C57BL/6 mice were administered Carprofen (5 mg/kg) subcutaneously 30 min before surgery, then anesthetized with isoflurane gas and kept warm on a 37 °C thermal pad. After shaving and disinfection with betadine, a mixture of 0.5% lidocaine (5 mg/ml) and 0.25% bupivacaine (2.5 mg/ml) was subcutaneously injected along the planned incision line. The spleen was exposed and the organoids were injected under the splenic capsule. After 10 min,

the spleen was resected by ligation. The wound was washed with sterile PBS, the peritoneal wall was closed with an absorbable polyglactin suture (Vicryl 4-0 or 5-0 coated), and the skin with wound clips. Buprenorphine (0.1 mg/kg) was injected subcutaneously during the wake-up phase. The animals were monitored until awake and in the days following surgery. The experiment was terminated 14 days after intrasplenic injection. All experimental procedures were performed in accordance with Swiss Guidelines and were approved by the Cantonal Veterinary Office Basel-City.

### Colon cancer organoids mixing species experiment
GFP+ mouse and GFP− human colon cancer organoids were mixed in a ratio of 1:1 and used for fragment-seq. Based on the GFP signal 144 wells were sorted with GFP+ mouse and 144 wells with GFP− human organoids.

**Cultivating human colon cancer organoids.** Human colon cancer organoids were obtained from the Visceral Surgery Research Laboratory led by Salvatore Piscuoglio at the University of Basel and cultured in Matrigel (Corning) as described above with the following adaptations to culture conditions: The culture medium (Advanced DMEM/F12, Life Technologies™) was supplemented with 10 mM HEPES (Life Technologies™), 2 mM Glutamax (Gibco), 1x B27 supplement (Life Technologies™), 10 mM Nicotinamide (Sigma Aldrich), 1.25 mM N-acetyl-l-cysteine (Sigma Aldrich), 500 nM A83-01 (Stemcell #100-0245), 50 ng/ml Human epidermal growth factor (hEGF) (Stemcell #78006.1), 100 ng/ml Recombinant human Noggin (Stemcell #78060), 100 ng/ml human R-spondin (LuBioScience #120-38-20), 10 nM Prostaglandin E2 (PGE2) (Stemcell #72634), 10 μM SB202190 (Stemcell #72634), 10 μM Y-27632 dihydrochloride (Rock inhibitor) (Stemcell #72304) and 10 nM Gastrin (Sigma Aldrich, #G9145).

**Cultivating mouse colon cancer organoids.** VilCreERT2;APCfl/fl;Tp53fl/fl;KrasG12D/wt (AKP) organoids obtained from Owen Sansom were labeled with the plasmid pMSCV-loxp-dsRed-loxp-eGFP-Ruo-WPRE[50] and cultured as described in 'Mouse model liver metastasis' section with the addition of 100 μg/ml murine recombinant Noggin to the culture medium (LuBioScience, #250-38-250).

### Tissue collection
**Liver.** Mice were euthanized and the liver was perfused with PBS (pH 7.4, Gibco) at a flow rate of 2–3 mL/min over an insertion of the cannula in the inferior vena cava. After the liver was fully perfused the gallbladder was removed and the liver was collected in ice-cold PBS. For Molecular Cartography and Visium, a piece of the liver was embedded in O.C.T.™ compound (Tissue-Tek) and snapped frozen in a metal beaker filled with isopentane (Sigma Aldrich) in liquid nitrogen. Snap-frozen tissues were stored at −80 °C. Molecular Cartography and Visium were performed on samples from mice that were also included in fragment-seq data.

**Spleen.** Mice were euthanized, the spleen was removed and collected in ice-cold PBS.

**Crohn's biopsy.** Tissue was surgically resected and collected in MACS tissue storage solution (Miltenyi Biotec #130-100-008) on ice for around 4–6 h before further processing. Recruitment of patients was carried out by the University Hospital of Zurich in accordance with ethics regulations. Coded patient IDs were provided to the study team, and the study has no further information on patient characteristics to report.

**Colon cancer organoids.** Matrigel of domes was dissolved with ice-cold PBS and organoids were collected in ice-cold PBS after shaking on ice for 30 min to remove all residual matrigel.

### Liver digestion for conventional scRNA-seq
Collected livers were cut into small pieces and incubated at 37 °C 300 rpm in a buffer containing low glucose DMEM (1 g/L D-Glucose/L-Glutamine, Pyruvate, Gibco #31885-023), 15 mM HEPES (Gibco) and 32 μg/ml liberase™ (Sigma Aldrich #05401119001) and 1x TripLE (diluted from 10x TripLE) (Gibco). After dissociation cells were filtered through a 100 μm strainer (Corning) and spun down at 300×g for 10 min to collect cells. Cells were then used for scRNA-seq capture on the BD Rhapsody system.

### Fragment-sequencing workflow
**Partial dissociation into fragments.** Tissues were partially dissociated into fragments using the mechanical force of a scissor. Fragments were then first filtered with a 400 μm strainer (PluriSelect #43-50400-03) to remove larger fragments that would clog the biosorter. Then fragments were filtered a second time using a 40 μm strainer (PluriSelect #43-50040-51) to remove single cells. Thereby, after the solution was filtered through the 40 μm strainer, the strainer was turned and larger than 40 μm fragments were washed off the strainer and collected in a 50 ml falcon tube in PBS for the spleen. Liver fragments were collected and sorted in low glucose DMEM (1 g/L D-Glucose/L-Glutamine, Pyruvate, Gibco #31885-023). The same filtering approach was used for Crohn's disease biopsies with a lower size strainer of 200 μm and a higher size strainer of 500 μm resulting in a suspension of fragments between 200 and 500 μm. Solution (Advanced DMEM/F12, Life Technologies™) with fragments was then put into a Petri dish and fragments were manually picked under a stereomicroscope (Leica) using a P200 pipette and tips.

**Fragment sorting using a large fragment biosorter (Copas).** Fragments were sorted into 96-well plates (non-binding, v-shaped, Greiner bio-one #651901) filled with 30 μl dissociation medium (see single-cell dissociation section), one fragment per well using a large fragment biosorter (Copas) with a 1000 μm large nozzle. The gates were set to allow the sorting of fragments within specific size ranges between 200 and 400 μm. The size was defined with a linear model by acquiring time of flight (TOF) of standard-sized beads (Megabead NIST Traceable Particles (Polysciences): 60 μm (#64200-15), 125 μm (#64225-15) and 175 μm (#64235-15)). TOF on the x-axis and extinction of the y-axis measures were then predicted for sizes between 200 and 400 μm from the linear model to the gate and sorted accordingly. For the organoid species mixing experiment GFP+ mouse organoids were sorted using the 488 nm blue laser at a power of 50. Gates for positive ones were drawn against a GFP− organoid control with a similar organoid size. Solution with fragments was diluted to around 10 events per second for proper sorting with pure/no doublets sorting mode. Other settings were the following: Power 50 mW; gain 1.0; PMT Volts green 600, red 750; drop width 7 mS; sort delay 23 mS; sample cup pressure 0.42 psi; diverter pressure 2.5 psi; sheath flow rate 57%. The sheath flow solution was PBS (pH 7.4, Thermo Fisher). Fragments were sorted directly into dissociation buffers. Spleen dissociation buffer was PBS (pH 7.4, Gibco). Liver dissociation buffer was a mix of low glucose DMEM (1 g/L D-Glucose/L-Glutamine, Pyruvate, Gibco #31885-023), 15 mM HEPES (Gibco) and 32 μg/ml liberase™ (Sigma Aldrich #05401119001) and 1x TripLE (diluted from 10x TripLE) (Gibco). Colon cancer organoids were sorted into 30 μl of 1x TripLE (Gibco).

**Manual picking of fragments from Crohn's disease biopsies.** Fragment solution (200–500 μml) after partial dissociation was put into a Petri dish and placed under a stereo microscope (Leica). Using a P200 pipette and low retention tips fragments were manually picked one by one into 100 μl epithelial dissociation medium (HBSS(−Ca$^{2+}$−Mg$^{2+}$) (Sigma Aldrich), 10 mM HEPES (Gibco) and 5 mM EDTA (Lonza)).

**Testing of sorting purity of single fragments per well.** Liver tissue was partially dissociated and used for sorting fragments between 200 and 450 μm in diameter. Nine plates were sorted and used for imaging on a Leica Thunder Imaging System. Plate scans were then visually inspected to count wells with none, one, or multiple fragments.

**Single-cell dissociation.** Spleen fragments were dissociated by applying mechanical force by pipetting up and down around 50 times using a P20 multichannel pipette. The Crohn's biopsy fragments were incubated at 37 °C and 300 rpm for two times 15 min with vortexing of plates in between for 30 s. After incubation plates were spun at 400×g for 10 min at 4 °C and the supernatant was carefully removed afterward. Then 100 μl of digestion medium (HBSS(+Ca$^{2+}$, +Mg$^{2+}$) (Sigma Aldrich), 0.5 mg/ml DNAseI (Roche Diagnostics #10104159001), and 0.5 Collagenase from *Clostridium histolyticum* (Sigma Aldrich #C5138)) were added and incubated at 37 °C and 300 rpm for 30 min. After incubation enzyme activity was inactivated by adding 50 μl of 20 mM EDTA/PBS (Lonza, pH 7.4 Gibco) shaking at 300 rpm 37 °C incubating for 5 min. Then plates were spun at 4 °C 400 rpm for 10 min and supernatant except around 30 μl was removed after. Colon cancer organoids were incubated at 37 °C for 12 min at 300 rpm and liver fragments were incubated at 37 °C for 22 min at 300 rpm.

**Labeling of cells with fragment-specific barcodes.** For labeling of fragments, the MULTI-seq lipid hashing method was used[25]. Lipid anchor and co-anchor were obtained from Zev J. Gartner laboratory at the University of California San Francisco. We designed a set of 288 barcodes with a minimum hamming distance of 3 using the Bioconductor package DNAbarcodes[51] (Version 1.20.0), incorporated primer binding sequences from the MULTI-seq method[25] and ordered them with a purity of standard desalting (MULTI-seq primers are provided in the Supplementary Data Files 1 and 2). Anchor and barcodes were mixed beforehand in a volume of 20 μl in PBS in a concentration of 50 nM for spleen, liver, and organoids and 100 nM for Crohn's biopsies in a ratio of 1:1. Co-Anchors were also diluted beforehand in PBS in the same concentrations. After the dissociation of fragments, 20 μl of Anchor:Barcode mix was added to the wells and mixed for 30 s at 700 rpm on a thermomixer at 20 °C. Then the cells were incubated on ice for 5 min. After incubation, the diluted co-anchor is added and again mixed for 30 s at 700 rpm at 20 °C followed by incubation on ice for 5 min. To quench the binding of lipids to the cells 100 μl of 10% BSA (Sigma Alrich) in PBS were added, mixed for 30 s at 700 rpm, and incubated on ice for 5 min. After incubation, cells from all wells were pooled into FACS tubes (Falcon) and spun at 400×g for 10 min for Crohn's biopsy and spleen, 300×g for 10 min for liver, and 300×g for 5 min for colon cancer organoids. Cells were then washed at least 2–3 times with 1% BSA in PBS. For the last wash cells were transferred into a 1.5 ml DNA low-binding tube (1.5 ml, Eppendorf) and spun a last time to then resuspend cells in only around 50–100 μl for counting and quality check using a hemocytometer and trypan blue solution (0.4%, Thermo Fisher). Samples were processed further for scRNAseq if there were at least 10,000 cells and the viability of cells was at least 70%.

**Single-cell RNAseq library preparation for fragment-seq experiments using BD Rhapsody and MULTI-seq.** Whole transcriptome analysis (WTA) on MULTI-seq labeled cells was performed using the BD Rhapsody Single-Cell Analysis System (BD Biosciences). In total, 9 liver samples were processed of mice that were injected with colon cancer organoids, one sample of healthy mouse liver, one sample of mixed colon cancer organoids, and two samples of Crohn's disease biopsies. In addition, cells from two livers of mice that were injected with colon cancer organoids were used for two experiments of conventional scRNA-seq (only WTA, no MULTI-seq library preparation). For each sample, a BD Rhapsody cartridge was loaded with approximately 10,000 cells. Single-cell capture and cDNA synthesis using the Single Cell Capture and cDNA Synthesis kit (#633731, #633733, #633773) was done following the manufacturer's protocol (BD Biosciences). Libraries were then prepared using the BD Rhapsody WTA Amplification Kit (#633801) following instructions of the mRNA WTA Library Preparation Protocol (BD Biosciences). For the MULTI-seq library, first, the BD protocol of Sample Tag Library Preparation was followed until purification of Sample Tag PCR1 product with the difference of adding the MULTI-seq primer (sequence according to McGinnis and Patterson et al. [25]) in a concentration of 10 μM instead of Sample Tag PCR1 Primer (kit component number 91-1088, BD Biosciences) for Sample Tag PCR1 reaction. After purifying the Sample Tag PCR1 product, an indexing PCR was done following instructions of the MULTI-seq protocol where small RNA TrueSeq indexing primers (Illumina #15004197) were used for i7 and the Forward Primer (kit component number 91-1085) from the BD WTA kit was used for i5.

**Single-cell RNAseq library preparation of fragment-seq experiments using Chromium 10X and MULTI-seq.** Fragment-seq library construction for murine spleen samples was performed with Chromium 10X. Libraries were generated following the manufacturer's instructions from Chromium Next GEM Single Cell V(D)J Reagent Kits v1.1 protocol (kit component number PN-1000165). In short: cells were resuspended in 0.04% BSA and mixed with a Master mix containing reagents for reverse transcription. The cell suspension was then loaded in GemCode Single-cell Instrument (10X Genomics) together with GemCode Single-Cell 5' Gel Beads. Cells and beads were fused to generate single-cell Gel Bead-in-Emulsions (GEMs). Within GEMs, cells were lysed and RNA was reverse transcribed. After GEMs were broken and cDNA was cleaned up, using DynaBeads MyOne Silane Beads (Thermo Fisher #37002D) and SPRIselect beads (Beckman Coulter #B23318), cDNA was amplified and cleaned up using SPRIselect beads. Then amplified cDNA was enzymatically fragmented and indexed sequencing libraries were generated by the following steps: end repair, A-tailing, adapter ligation, post-ligation SPRIselect cleanup, and sample index PCR. For MULTI-seq library preparation instructions were followed from McGinnis and Patterson et al. [25]. In short: The MULTI-seq primer (according to MULTI-seq protocol instructions) was added in a concentration of 10 μm to the cDNA amplification mix. After cDNA amplification during SPRIselect clean up the non-bound fraction (containing small cDNA fragments) was saved and cleaned up with SPRIselect beads in a higher ratio to enrich for small MULTIseq barcodes. These products were then used for index PCR using the SI-PCR Primer from the 10X kit for the i5 and one of the small RNA TrueSeq index primers for the i7 (Illumina #15004197).

## Visium library preparation

Two mouse liver samples with visible micro-metastasis were processed. A 10X Visium Spatial Gene expression slide was put into the cryostat (Leica CM3050S) to calibrate its temperature to −20 °C. Then 10 μm sections of metastatic mouse liver samples were cut and placed within the capture area. The capture slide was then stored in a slide container at −80 °C until the next day for further processing. cDNA libraries were generated following the manufacturer's instructions. In short: Tissues were fixed with methanol and hematoxylin and eosin (H&E) staining was done to check tissue quality and morphology. Then tissue lysis, reverse transcription, second strand synthesis, and cDNA denaturation were performed on the slides. Permeabilization time of 10 min was assessed beforehand with the Tissue Optimization Protocol. Reactions were transferred into PCR tubes and qPCR was done to measure cDNA concentration. cDNA was then amplified by PCR using cycle numbers defined by qPCR. Final library preparation steps (End repair, A-tailing, adapter ligation, and sample index PCR) were done to generate indexed sequencing libraries.

## Quality assessment of libraries and sequencing

Quality and quantity of all libraries were assessed using the dsDNA high-sensitivity (HS) kit (Life Technologies #Q32854) on a Qubit 4 fluorometer (Thermo Fisher) and high-sensitivity D1000 reagents and tapes (Agilent #5067-5585, #5067-5584) or high sensitivity D5000 reagents and tapes (Agilent #5067-5593, #5067-5592) on a TapeStation 4200 system (Agilent Technologies). Paired-cell sequencing was performed for all libraries (WTA BD libraries: read 1: 60 bp, index read: 8 bp, read2: 62–100 bp; WTA 10X libraries: read 1:26 bp, index read: 8 bp, read2: 88–96 bp; MULTI-seq libraries: read 1:26 bp (10X), 60 bp (BD Rhapsody), index read: 6 bp, read2: 62-100 bp)) on a NovaSeq 6000 system (Illumina) using NovaSeq SP Reagent Kits (100 cycles) v1.5 and S4 Reagent kits (200 cycles) v1.5 with XP workflow. WTA libraries of BD Rhapsody and 10X were sequenced at 50,000 reads/cell, MULTI-seq libraries of both 5000 reads/cell, and Visium libraries at 50,000 reads/spot.

## Single-cell RNAseq and Visium data preprocessing

**Demultiplexing.** BCL files were demultiplexed using Bcl2fastq v2.20.0.422 from Illumina to convert them to FASTQ files.

**BD Rhapsody data was preprocessed using zUMIs.** FASTQ files from WTA libraries of BD Rhapsody data from fragment-seq and conventional scRNA-seq were processed using the zUMIs[52] (v2.9.4) platform to convert reads to count matrices per cell. For gene alignment STAR[53] (v2.5.2b) was used with the following gene codes: for human Crohn's disease biopsy samples GRCh38 v2020-A; for mouse liver samples GRCm38 vM25 fused with GFP 3′ UTR sequence and for the colon cancer organoids mixing species experiment a fused index of GRCm38 v2020-A, GRCh38 v2020-A, and GFP 3′ UTR sequence. Three liver samples (S1−3) were sequenced twice to acquire deeper sequencing, fastq files for read1 and read2 were merged and the merged files were used as zUMI input.

**10X data was pre-processed using Cell Ranger.** Mouse spleen data single cell count matrices were generated using Cell Ranger (v5.0.0) (10X Genomics) with GRCm38 v2020-A gene code.

**Visium data was pre-processed using Space Ranger.** Mouse liver Visium data was pre-processed using Space Ranger (v1.2.0) (10X Genomics) with GRCm38 v2020-A gene code.

## Fragment-sequencing downstream analysis

Downstream analysis of UMI count matrices was done in R version 4.1.0 and most analysis was done using the following packages: Seurat[54] (v4.0.3), scran[55] (v1.22.1), and SingleCellExperiment[56] (v1.16.0). Dplyr[57] (v1.0.7) and tidyverse[58] (v1.3.1) were used for data wrangling. Plotting was mostly done with ggplot2[59] (v3.3.5).

**Conversion of gene code numbers from zUMI outputs to gene names.** Genecode numbers were converted using the biomaRt[60,61] Bioconductor package (v2.50.3) with musmusculus_gene_ensemble version 95 for mouse data and hsapiens_gene_ensembl version 95 for human data. For the organoid species mixing experiment both gene codes were used. GFP gene name was included.

**Fragment barcode classification and integration with whole transcriptome analysis (WTA).** For allocation of MULTI-seq barcodes to single cells the deMULTIplex[25] (v1.0.2) workflow on the GitHub repository https://github.com/chris-mcginnis-ucsf/MULTI-seq was followed. Briefly, a sample barcode UMI matrix per cell was generated. Then cells were assigned to specific barcodes following the classification workflow. Cell barcodes of classified cells were then matched with cell barcodes of WTA. This resulted in positive, negative, or doublet-classified cells.

**Pseudobulk clustering with spatially defined marker genes to locate fragment position.** Differential gene expression analysis on Molecular Cartography data between proximal and distal areas was done using the Seurat function 'FindAllMarker' genes with default parameters. Only the following cell types were considered because they were also sufficiently present in fragment-seq data: B cells, Kupffer cells, LECs, metastatic cells, macrophages/monocytes, and T cells. Pseudobulks of fragments from one metastatic murine liver sample using the 'AverageExpression' function from the Seurat package. The top 10 differentially expressed genes per group (distal or proximal) were then used to cluster fragments using 10 principal components (PCs) and a resolution of 0.4. Proportions of broad cell types were then assessed between cluster 1 and cluster 2.

## Mouse liver-specific fragment-seq downstream analysis

**Normalization and batch effect correction.** All 10 liver samples were merged after the generation of Seurat objects and UMI (unique molecular identifier) counts underwent SCT normalization (Seurat function, 'SCTransform') which normalizes and scales data using Pearson residuals and finds variable features. Then the merged Seurat object was transformed into a SingleCellExperiment object and batch effect correction was done using MNN (mutual nearest neighbors) correction within the batchelor[62] Bioconductor package (v1.10.0).

**Quality control and clustering.** After batch effect correction low-quality cells (lower than 200 features and higher than 20% of reads mapped to mitochondrial genes) and doublets (higher than 7500 features) were removed and 10 MNN corrected principal components (PCs) were used for clustering of cells in uniform matrix approximation and projection (UMAP) two-dimensional space.

**Cell type annotation.** Initial clustering with 10 MNN corrected PCs and a resolution of 1 was first broadly annotated using cell type markers from the liver cell atlas[22]. Then each broadly annotated cell cluster was further investigated for subtypes using the Seurat sub clustering function 'FindSubCluster'. Subclusters were then annotated with the help of the liver cell atlas and investigation of differentially expressed genes (DGE) using the Seurat function 'FindAllMarkers' with a non-parametric Wilcoxon Rank Sum test with default parameters (min.pct 0.25 and logfc.threshold 0.25).

**Integration of fragment size from biosorter data.** A standard curve was generated with TOF measurements of standard-sized beads. TOF measurements of sorted fragments were then fitted into the linear model to calculate the size in diameter.

**Cells per fragment cutoff.** For further analysis, only fragments with at least five cells were used.

**Lobule layer classification.** For each fragment, a zonation coordinate (ZC) based on zonated landmark genes in LECs was calculated following a previously developed approach[63]. In detail, we first generated pseudo bulks from the LECs of each fragment. Next, we compared this to central and portal landmark genes from Halpern et al. [15], which detailed mean gene expression in the 9 different lobule layers of hepatocytes. The expression of genes in *Halpern et al.* was normalized above $10^{-5}$ across all layers and had a fold change between L1 and L10 of at least 10% with an average ratio between standard arrow and mean over all layers of <0.2 (which was implemented to discard highly varying genes)[63]. Next, genes expressed in fragment-pseudo-bulks were normalized by dividing their expression by the maximum level of expression across fragments to ensure equal contribution of all genes. GThen the sum of central and portal landmark genes (cLM and pLM) was calculated for each fragment and used to calculate ZCs by the following calculation: $ZC = pLM/(pLM + cLM)$. In the end, ZCs were

rescaled so that 0 is the most central and 1 is the most portal coordinate: $ZC = (ZC-min(ZC))/(max(ZC)-min(ZC))$. Fragments without LECs were removed from further analysis at this point. ZCs of fragments were then grouped into lobule layers L1-L10 (L1: $ZC < 0.1$, L2: $ZC < 0.2$, L3: $ZC < 0.3$,..., L10: $0.9 < ZC \leq 1.0$) There were not many fragments from the most central and most portal areas so they were grouped into L1-L3 and L8-L10. To control the zonation algorithm in our data, landmark genes in hepatocytes were analyzed from fragments grouped into central (L1-L5) and portal (L6-L10) veins.

**Splitting of the dataset for different analyses.** After pre-processing analysis data was split into three datasets. The healthy liver sample ($n = 1$), the liver samples of mice that were injected with colon cancer organoids ($n = 9$) for analyzing liver zonation during metastasis formation, and liver samples with a high amount of metastatic cells (at least 20 within all cells) ($n = 3$, samples S3, S6 and S7) for analysis of different metastatic niches (Supplementary Fig. 6a).

**Metastatic distance classification.** Fragments with metastatic cells were grouped into 'proximal' and fragments without metastatic cells into 'distal' categories

**Analysis of zonation-specific genes.** Differential gene expression analysis was performed using edgeR[64–66] (v3.36.0). For this single-cell counts were summed across fragments with at least 5 cells of a cell type of interest to derive a single expression vector per fragment. After removing lowly expressed genes using the 'filterByExpr' function in edgeR, a negative binomial generalized log-linear model was fitted to the remaining genes with the lobule layer as an ordered factor covariate (L1-L3 < L4 < L5 < L6 < L7 < L8-L10). The linear coefficients were then used for fitting and the sample names were used as a blocking factor to account for batch effects. The 'glmQLFTest' function was used to identify genes with coefficients for the linearly encoded factor significantly different from 0 at a Benjamini–Hochberg adjusted p-value of 0.05. Degrees of freedom and other statistical properties can be obtained from source code[67]. This analysis was done for LECs and KCs. In LECs only genes that were not in the landmark gene panel to calculate ZCs were further investigated.

**Differential gene expression (DGE) analysis between two groups.** DGE analysis was done as described above with two differences. First, single-cell counts were summed across fragments with at least two cells of a cell type of interest, and second, instead of using lobule layers as an ordered factor covariate, two groups were used as factor covariates.

**Analysis of differences in cell type abundance between two groups.** To identify changes in cell-type-specific abundance between two groups, normalized log counts of cluster abundance were computed using the 'cpm' function in edgeR[64–66] (v3.36.0) accounting for the total number of cells per sample[68]. After specifying a design matrix with group labels (veins 'central' and 'portal' or distances to metastasis 'proximal' and 'distal') as covariates and sample names as blocking factors, the dispersion parameter of the negative binomial model was estimated using the 'estimateDisp' function in edgeR with trend = 'none'. a negative binomial generalized log-linear model was fitted with 'glmQLFit' function (robust = TRUE, abundance.trend = FALSE) for each cell type. The 'glmQLFTest' function was then used to identify cell types with coefficients significantly different from 0 at a Benjamini–Hochberg adjusted p-value of 0.05.

**Ligand–receptor (L–R) interaction between different groups using CellPhoneDB.** L–R interaction analysis was done using the Python Package CellPhoneDB[30] (v4.0.0) following instructions on the GitHub repository https://github.com/ventolab/CellPhoneDB. In short, gene expression of annotated clusters was used as input to match with known L–R interaction pairs from the CellPhoneDB public repository using default parameters. The average ligand and receptor expression between two cell types were represented by the mean values which were calculated using the percentage of cells within a cluster expressing the ligand or receptor and their gene expression mean. A null distribution of means for randomly permuted annotated cluster labels was then used to determine p-values. Analysis was done for separate groups of veins (central and portal) and distances to metastasis (proximal and distal). Differences in L–R interaction scores between the two groups were then visualized in a barplot with L–R pairs ordered decreasing by the difference in interaction scores between the two groups.

**Assessment of bias between different fragment sizes and cell number cutoffs.** The differences between fragment size and cell counts of fragments between two groups (pericentral and periportal, distal and proximal) were assessed with a non-parametric Wilcoxon signed-rank test using the 'ggsignif 'function from ggplot2[59] R package. Fragments were then grouped into two different size ranges (211–325 and 326–457 μm) to test for different fragment sizes. For testing the influence of different cell numbers an object with fragments having at least 5 cells/fragment was compared to an object with fragments with at least 20 cells. Differences in cell type proportions were assessed by umap plotting split by different scenarios. DEGs of KCs and LECs were assessed between two groups as described previously with batch and lobule layers as blocking factors. Zonated genes were assessed in LECs from different scenarios described previously and plotted in Volcano plots (for size ranges or cellular cutoff). L–R interaction analysis of datasets with different cutoffs was performed as described in the section *Ligand–receptor (L–R) interaction between different groups using CellPhoneDB*.

**Ligand–receptor (L–R) interaction between different groups using CellPhoneDB.** Interaction scores of datasets with different sizes and cell number cutoffs were then plotted in bar plots for comparison. To assess the influence of fragment size or sample size on L–R interaction analysis, cells from the pericentral and periportal datasets which included the whole size range (211–457 μm) were randomly downsampled to the same amounts of cells as present in the pericentral and periportal areas of the small fragment-size data sets (211–325 μm).

### Conventional scRNA-seq downstream analysis
Cell count matrix generation, gene name conversion, clustering, and annotation were performed as described in previous sections for fragment-seq downstream analysis. The normalized, log-transformed counts were then used to map the data onto the fragment-seq dataset using the 'fastMNN' function using the top 6000 HVGs as implemented in the batchelor package (v1.10.0)[62]. The first ten principal components from the batch-corrected PCA space were then used to compute the UMAP in Fig. 1. Quality features (median UMI counts, gene features, and ratio of mitochondrial to cytoplasmic genes) were assessed after accounting for different read depths by downsampling and only considering cells with 30,000 reads. Therefore we re-run zUMIs[52] (v2.9.4) for all samples using '30000' as a counting_opts downsampling parameter.

### Colon cancer organoids mixing species-specific downstream analysis
**Integration of GFP fluorescent signal from biosorter data.** Fragment size was calculated with the help of a standard curve from standard-sized beads. The GFP signal was then normalized by dividing it by the fragment size to account for autofluorescence that is higher in larger fragments.

**Quality control, normalization, clustering, and annotation.** Low-quality cells were removed with lower than 200 features and larger than 30% of reads mapped to mitochondrial genes. UMI counts were then normalized and scaled using the 'SCTransform' Seurat function and 10 PCs were used for clustering in UMAP space. Cell clusters were then annotated as human or mouse depending on the species of genes being expressed. Based on the knowledge of sorting, cells could also be annotated by their species as well.

**DecontX to remove cell-free RNA.** During data exploration of human and mouse UMI reads per cell we found a lot of cell-free RNA in fragment-seq data even after the removal of low-quality cells. This was probably due to the low quality of colon cancer organoids (~20–30% dead cells before single-cell capture). Therefore decontamination of data was done using the decontX function (default parameters) within the celda[69] R package (v1.12.0).

**Analyzing the fraction of correctly and wrongly assigned cells.** There are two annotations, the cell species annotation and the well annotation that are established by sorting a human or mouse organoid. By matching these two pieces of information the proportion of wrongly and correctly assigned cells for each fragment could be analyzed, wrong if there were mouse cells in human wells and human cells in mouse wells, correct if the species was matching. Three fragments were found to be 100% made out of wrongly assigned cells and were therefore allocated to the opposite species well because these are most probably due to a fluorescence sorting error.

### Mouse spleen-specific downstream analysis
**Quality control, normalization, clustering, and annotation.** Two samples were merged after fragment barcode integration; no batch effect correction was needed. Low-quality cells were removed, with lower than 200 features and larger than 10% of reads mapped to mitochondrial genes, and doublets were removed with larger than 6000 features. UMI counts were then normalized and scaled using the 'SCTransform' Seurat function and 10 PCs were used for clustering in UMAP space. Cell clusters were annotated using cell type marker genes from Medaglia et al. [10]. Only fragments with at least 5 cells were considered for plotting.

### Crohn's disease biopsy-specific downstream analysis
**Quality control, normalization, clustering, and annotation.** Two samples were merged without the need for batch effect correction. After low-quality cells and doublets (cells with lower than 200 features, larger than 25% of mitochondrial reads, and larger than 5000 features) were removed, UMI counts were normalized and scaled ('SCTransform') and 10 PCs were used for clustering. Cells were annotated using marker genes from Martin et al. [70]. At least 5 cells per fragment were required for plotting.

### Visium data downstream analysis
**Normalization, clustering, and spatial area annotation.** Samples were processed separately, normalization and scaling were done using the 'SCTransform' Seurat function and clustering was done using 10 PCs in UMAP space. Clusters were then annotated into the following areas: portal and central veins (also considered as 'distal' to metastatic sites), and metastasis (considered as 'proximal' to metastatic sites) based on landmark genes.

**Comparison of the number of gene features.** Mean values of the number of gene features between proximal and distal metastatic areas of Visium data were compared with mean gene feature values of both areas from fragment-seq data. The same two metastatic liver samples were used.

**Batch effect correction.** The two Visium samples were merged and batch effect correction was done using MNN (mutual nearest neighbors) correction within the batchelor[62] Bioconductor package (v1.10.0).

**Deconvolution.** For the deconvolution of spots, annotated fragment-seq data was used as a reference. The top 20 genes per cell type cluster were selected that were also expressed in Visium datasets. These genes were used for deconvolution using the SCDC[71] package (v0.0.0.9000). Spots were allocated to a specific cell type if at least 75% of genes could be assigned to one cell type. Spots with <75% are annotated as mixed.

**Public Visium data analysis.** Public Visium data from Guilliams et al.[22] was used from wild-type mouse and NAFLD (non-alcoholic fatty liver disease) mouse models. Spots annotated for different liver zones (central, mid, periportal, and portal) were grouped and tested for gene expression of zonation-specific genes.

### Highly multiplexed FISH (Molecular Cartography™)
**Sample preparation, probe design, Imaging, and pre-processing.** These steps were done as previously described[22]. In brief, for sample preparation, liver samples (4 mouse liver metastasis samples of which two samples had visible micro-metastasis) were frozen and sectioned into 10 μm slices as described for Visium; sections were placed within capture areas on Resolve BioScience slides. Afterward, slides were sent to Resolve BioSciences on dry ice, where they were processed further. Samples were fixed and underwent 100-plex combinatorial single-molecule fluorescence in-situ hybridization. During multiple cycles of color development, imaging, and decolorization a unique combinatorial code for each target gene was generated. 100 genes were chosen from fragment-seq analysis (20 genes were chosen to define cell types and 80 genes were hits to validate) and their probes were designed using Resolve's proprietary design algorithm that makes sure that probes are specific with little off-target binding. Fluorescent signals were imaged on a Zeiss Celldiscoverer 7 microscope with a final magnification of ×25. Each region underwent 9 imaging rounds and 16 z-stacks were acquired. Java and C++ scripts were then used for spot segmentation and images were pre-processed to remove background fluorescence. Raw data images from different imaging rounds were aligned during which images had to be corrected using an iterative closest point cloud algorithm. Then a profile for each pixel was created using the information of 16 values (16 images from two color channels in 8 imaging rounds).

**Downstream analysis.** Image analysis was performed in ImageJ using genexyz Polylux tool plugin from Resolve BioSciences.

**Cell segmentation using Cellpose.** Cellpose[72] (v.2.0.4) was used to segment nuclei from the DAPI images with the pre-trained nuclei model and flow_treshold 0.5, cellprob_threshold −0.2. The nuclear segments were then expanded by 10 pixels (1.38 μm) using the 'expand_labels' function implemented in scikit-image and transcripts were subsequently assigned to the expanded segments. Segments larger than 4 median absolute deviations (MAD) plus the median segment area were removed from the analysis. During clustering and cell type annotation, low-quality clusters of cells were removed which could not be properly annotated.

**Normalization, clustering, and annotation.** Count matrices of segmented cells were normalized and scaled using the 'SCTransform' function of Seurat and then 10 PCs were used for clustering in UMAP space. Cell clusters were then annotated using marker genes from fragment-seq analysis; cells that could not be properly annotated were removed from the analysis. Annotation was then projected on cells as an overlay on Molecular Cartography DAPI images in ImageJ.

**Feature area integration.** Signals for landmark genes defining central (*Cyp2e1*), portal (*Cyp2f2*), and metastatic areas (*Gpx2*) were used to visualize different areas, then areas were manually drawn and their *x* and *y* coordinates were exported. Coordinates from different areas of the image were then matched with the *x* and *y* coordinates of cell segmented count data to annotate single cells by their area of origin.

**Differential gene expression (DGE) analysis between two groups.** DGE analysis was done as described for fragment-seq analysis. But instead of using the sum of single cell counts across fragments, counts were summed across spatial feature areas.

**Analysis of differences in cell type abundance between two groups.** Cell-type proportions were analyzed as described for fragment-seq analysis.

**Colocalization analysis.** A spatial neighborhood graph was constructed based on the Euclidean distance in 2D space of the centroids of the segmented areas. In this graph, vertices represent the cells that are connected by an edge if the distance is smaller than 10 μm. To construct the graph, we utilized a kd-tree-based nearest neighbor search in a pre-defined radius of 10 μm as implemented in the R function 'nn2' (RANN v.2.6.1,searchtype = 'radius') with a sufficiently large *k* (*k* = 41). This approach runs in $O(M \log M)$ time and avoids the computation of the distance matrix for thousands of objects. The resulting adjacency matrix was then used to construct a graph using the igraph[73] package (v.1.3.4). From this graph the number of edges between cell types was computed in each region of interest (ROI) and divided by the sum of the number of cells for each cell type pair to normalize for total cell numbers in the annotated regions. For each slide, the difference in the normalized number of edges between the two groups of ROIs, e.g. proximal versus distal, was subsequently computed. This value was compared to an empirical null distribution derived from randomly permuting the labels of the vertices (*m* = 1000) per slide. This approach takes tissue composition and spatial structure into account and allows the computation of *p*-values as $p = (b + 1)/(m + 1)$ where *b* is the number of times the permutation produced a higher number of edges between two cell types than observed and *m* the total number of permutations[74]. This was done for each slide and possible cell–cell interactions to derive a score that represents the fraction of images in which a specific interaction was significant, with the sign representing co-localization or avoidance; visualization was adopted from ref. 75.

**Comparison of cell type abundances between fragment-seq and Molecular Cartography.** Only cell types that could be robustly detected in Molecular Cartography, because of the presence of cell type marker genes in the gene panel, were included in this analysis. Cell-type proportions from both protocols were calculated and plotted in a barplot.

**Immunofluorescence (IF) stainings**
Mouse metastatic liver two weeks after intrasplenic injection with CRC organoids were fixed with formalin and embedded in paraffin. Using a microtome (Leica) 5 μm sections were cut and placed on slides. Slides were then left to dry overnight. The next day tissues were deparaffinized and rehydrated using consecutive incubations as follows: 15 min xylene (Sigma Aldrich) solution 1, 15 min xylene solution 2, 15 min xylene solution 3, 5 min xylene/ethanol, 2 min 100% ethanol (Sigma Aldrich), 2 min 100% ethanol, 2 min 95% ethanol, 2 min 95% ethanol, 2 min 70% ethanol, 2 min 70% ethanol, 2 min $H_2O$, 2 min $H_2O$. After, antigen retrieval was performed by incubating the slides for 30 min in a low pH buffer (10 mN sodium citrate solution, pH 6.0, Sigma Aldrich). Then slides were washed with PBS two times for 5 min and incubated with 10% BSA/PBS 0.25% Triton X-100 solution (Sigma

Alrich) at room temperature for blocking. Then slides were incubated with primary antibodies (rabbit anti-mouse SPP1: Catalog #MAB808, Clone # 2139B, Lucerna-Chem AG, validated through western blotting by manufacturer, working concentration 5 μg/ml; goat anti-mouse VCAM1: Catalog #AF643, polyclonal, Lucerna-Chem AG, validated through western blotting by manufacturer, working concentration 5 μg/ml) at 4 °C overnight. The next day, slides were washed 2 times with PBS 0.25% Triton X-100 for 5 min before incubation with the secondary antibody (Cross-absorbed donkey anti-goat Alexa Fluor 647, #A21447, polyclonal, Invitrogen, working concentration 1:400; cross-absorbed goat anti-rabbit AF647, polyclonal, #A21244, Invitrogen, working concentration 1:400) at room temperature for 1 hour. After incubation slides were washed 2 items with PBS 0.25% Triton X-100 following incubation with DAPI (Thermo Fisher) in PBS (0.05 μg/ml) for 10 min at room temperature. Then slides were washed 2 times with PBS before mounting them with ProLong Gold antifade reagent (Invitrogen). Slides were then images with a Leica immunofluorescent microscope.

**Schematic drawings**
Schematics for experimental procedures or overviews have been created with BioRender.com.

**Statistics and reproducibility**
No statistical method was used to predetermine sample size, instead, sample sizes were chosen based on comparable studies in the field (for example paired cell sequencing[15] *n* = 3, PIC-seq *n* = 4–7[16], Clump-seq *n* = 3[17]). In addition, biological interpretations were constrained by statistical significance obtained through the given sample size. The experiments or experimental animals were not randomized. The investigators were not blinded to allocation during experiments and outcome assessment. Fragments with <5 cells were filtered out during analyses, as they are not sufficient to represent the cellular heterogeneity of a niche. In addition, for analyses centered around distal and proximal fragments we focused on the 3 out of 9 metastases-bearing animals with the highest tumor burden (at least 20 metastatic cells per sample). Likewise, to assess the differentially expressed genes (DEGs) of macrophages/monocytes between distal and proximal regions from MC we used only 2 of the 4 samples from mice with high metastatic burden (visible macrometastases). Sex was not considered a factor in study design, as the fragment-seq method has no gender-specific components. Both male and female mice were used. The fragment-sequencing method was reproducible across samples and independent experiments. Where applicable, statistical tests are indicated in the figure legends.

**Reporting summary**
Further information on research design is available in the Nature Portfolio Reporting Summary linked to this article.

# Data availability
The data generated in this study (gene expression data for human fragment-seq, RNA-sequencing data for murine scRNA-seq and fragment-seq, and Visium data) have been deposited in the Gene Expression Omnibus (GEO) database with accession number GSE216189. The Molecular Cartography data from Resolve are deposited at Zenodo with the accession number 8413573[76]. The raw RNA-sequencing data for Crohns' Disease biopsies are not available neither publicly nor by request as consent for the publication or sharing of this type of data was not obtained. The processed gene expression data from patients are included in the GSE216189 dataset. For non-sensitive data types, GSE216189 includes raw the sequencing data. Data generate in this study on fragment sizes, UMI counts, GFP signal and cell counts of species mixing experiments, cell type proportion in fragments, enriched genes in different hepatic zones and/or cell types, L-R

interactions, colocalization scores, fragments sorted per cell, TOF of beads, UMI counts, gene counts and percentage of mitochondrial genes in scRNA-seq and fragment-seq, analyzes with different fragment size cutoffs, fragment size and cellularity, number of metastatic cells per sample and fragment, cell type enrichment in fragment-seq, MC, and Visium as well as Visium gene counts are provided in the Source Data file. The previously published data used in this study from the liver cell atlas[22] are available in the GEO database under accession number GSE192742. Source data are provided with this paper.

## Code availability

The code generated in this study is available at: https://github.com/ Moors-Code/Fragment-sequencing and is citable with the following https://doi.org/10.5281/zenodo.8246953[67].

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

## Acknowledgements

We thank Owen Sansom (Beatson Institute for Cancer Research, Glasgow, UK) for the AKPS organoids. We also thank Michael Scharl and Barbara Maria Szczerba (Gastroenterology and Hepatology, University Hospital Zürich, Switzerland) for the collection of Crohn's disease biopsies and we thank the patients and their families for participating in this study. For help and support with the large fragment biosorter, we thank Sven Panke and Steven Schmitt (Department of Biosystems Science and Engineering, ETH, Basel, Switzerland) and Erich Brunner (Department of Molecular Life Sciences, University of Zurich, Switzerland). We thank Laura De Vargas Roditi (Department of Biosystems Science and Engineering, ETH, Basel, Switzerland) for interesting discussions for analysis pipeline development. And finally, we thank Zev J. Gartner (UCSF, San Francisco, USA) for providing us with lipid anchor and co-anchor (MULTI-seq) and Christopher S. McGinnis (UCSF, San Francisco, USA) for technical advice. We used BioRender for creating schematic drawings. This work was funded by the Swiss National Science Foundation (PCEFP3_181249 to A.E.M.) and the Leona M. and Harry B. Helmsley Charitable Trust (Gut Cell Atlas grant to A.E.M.).

## Author contributions

K.H. and C.B. performed the experiments. K.H., K.B., and I.E.A. performed the data analysis. S.P. provided essential reagents. A.M. supervised the study. K.H., X.F., and A.M. wrote the paper. All authors discussed the results and commented on the manuscript.

## Competing interests

The authors declare no competing interests.
