## [Peer Review File · Nature Communications]

Fragment-sequencing unveils local tissue microenvironments at single-cell resolutionEditorial Note: This manuscript has been previously reviewed at another journal that is not operating a transparent peer review scheme. This document only contains reviewer comments and rebuttal letters for versions considered at *Nature Communications*.

REVIEWER COMMENTS

Reviewer #1 (Remarks to the Author):

We thank to the author for providing clarification and additional data on the reviewer concerns.

However, a few major concerns remain unresolved.

1. The z-score and other data normalization method comparisons for heatmaps should be provided. and how would other normalization methods affect the clusters visualized? Seems like only 1 method provided them what the authors wanted to show. This means that data mining should be robust
2. The sphere-seq is an unacceptable name because nothing in the dissociated blobs resembles spheres. As a diligence, I find reviewer concerns were not taken into account when responding. We would highly recommend renaming this technique to avoid future confusions.
3. The spatial location of where spheres came from the tissues is also very crucial. This technique is only impactful if the users can track the position of the blobs. Thus, the readers (and of course reviewers) would be interested in a proof-of-concept of the proposed approaches as described in the paper. Putting such an important direction out-of-scope is premature from a revision perspective. This is especially needed because of direct comparisons of MERFISH, DBIT-seq etc. technologies to the proposed method. They are only fair comparisons if the spatial data is retained, measured, and incorporated into this technology.

Reviewer #2 (Remarks to the Author):

We applaud the revised manuscript by the authors that much more rigorously benchmark sphere-seq based on this reviewer's feedback. There remain a few minor comments that should be addressed before this manuscript is acceptable for publication, and better inform potential users for this methodology.

Related to Q1.

The authors' response to the concern regarding the variation of the sphere size, found them "to be comparable in size and cell number (Supplementary Fig. 6a,b and Supplementary Fig. 7c,d)". The boxplot representation here should be replaced with a violin plot, as it is difficult to visually confirm that. Similarly, the term "comparable" should be toned down since there are differences observed.

It appears from Supplementary Fig 6g that several L-R interactions were not detectable when split across the two different size groups? Perhaps this is one of the limitations of the method caused by the uncontrollable size difference. That authors should comment on this in the results and discussion.

Related to Q3.

The authors did not fully address my concern related to any potential mechanical fragmentation induced transcriptional changes beyond those from scRNA-seq. The comparison in Q4 thereafter has more to do with cell type annotation and transcript counts rather than additional mechanically induced transcriptional responses due to sphere-seq. This warrants some form of attempt at showing little/no changes along with discussions.

Related to Q4.

The authors now present Supplementary Fig 3d, which indicates NS differences of UMI and gene counts, mito/cyto ratio of genes. There seems to be a considerable difference between the methods based on the mean values (e.g. 500 vs >1000 UMI per cell; 200 vs ~300 gene counts per cell). This NS is almost certainly due to the low sample size of $n=2$ in scRNA-seq vs $n = 9$ in sphere-seq. If sample sources is really the limiting factor, one can potentially subsample or bootstrap the data to better compare their average UMI/gene counts or ratio, and better test for statistical significance.

The second question is that it seems based on these values alone, sphere-seq is better than scRNA-seq. Can the author potentially explain why it would be better, as for the protocol cells likely go through more perturbation etc, or is it due to differences in sequencing depth etc that should be accounted for in the data presented above.

For Rev Fig2, the authors show pearsons correlation scatter plots. I assume each dot is a gene, and the x, y axis is the average (?) normalized count for that gene in scRNA-seq and sphere-seq. Are the genes plotted here all the genes? What does this plot look like if we select variable genes in scRNA-seq + variable genes in sphere-seq, and plot them in such scatter plots? These may in part be related to my concerns in Q3 related to potential mechanically induced transcriptional changes added on by sphere-seq.

Related to Q5.

The Supplementary fig 9e would be better serviced if it is limited to cell types that can be robustly detected in both sphere-seq and Molecular Cartography for a more meaningful comparison

Related to Q6.

The authors raised a very good point that should be included in the discussion section.

Related to Minor point Q4.

Boxplots are completely and statistically meaningless with 3 points. The upper and low quartile in there, with 3 points, do not mean anything.

As previously mentioned, the authors should replace them, even with just colored dots, or a single line indicating the mean. If the rationale is optical consistency, then one can color the dots without the need for the meaningless boxplots.

Related to Reviewer 3 Q2:

The authors mentioned not using individual spheres for L-R analysis but instead grouped them together. This confounds this reviewer, as it seems like this defeats the purpose of sphere-seq

Additional comments:

The authors should discuss the following to better help readers understand the potential for the method:

1. Whether this protocol can be eventually be potentially used on freshly acquired human tissue, and some of its potential future applications.
2. For precious and limited samples, would the high cell loss rate e.g. mentioned in Reviewer3 Q1 be a limitation?
3. Are there alternative ways that can be used to group different spheres, eg a sphere with 5 cells but with 1 cancer cell is certainly different from a sphere with 50 cells but with 1 cancer cell.

Reviewer #3 (Remarks to the Author):

The authors have satisfactorily addressed my comments/concerns through additional experiments, analyses, comparisons with published data, modifications to the text, and visual improvements to the figures. I do not have any additional major comments. I have one minor comment that I hope the authors can respond to, but I do not feel that publication of this manuscript is conditional on the response.

In reviewer figure 3, the authors report that only 50% of their single cells were kept due to Multi-seq classification. As an increase in cell number recovery would improve proportional and other analyses of Sphere-seq data, did the authors attempt the negative-cell reclassification for their data (<https://github.com/chris-mcginnis-ucsf/MULTI-seq>)? Relatedly, for the cells that were negative, was there a cell type bias relative to the positive cells? In other words, did the Multi-seq process work better for some cell types than others?

We are grateful to the referees for their detailed second evaluation of our manuscript. Below we detail how we have addressed each point (the reviewer's comments are in black, and our responses are in blue). For your convenience, we have also included the resulting changes in the manuscript in purple.

REVIEWER COMMENTS

Reviewer #1 (Remarks to the Author):

We thank to the author for providing clarification and additional data on the reviewer concerns.

However, a few major concerns remain unresolved.

1. The z-score and other data normalization method comparisons for heatmaps should be provided. and how would other normalization methods affect the clusters visualized? Seems like only 1 method provided them what the authors wanted to show. This means that data mining should be robust

We thank the reviewer for the comment regarding data normalization. It appears that in the process of the discussion, some concepts have been confused and we would like to briefly clarify. Z-score and normalization represent two different concepts: For clarification, z-scores are computed row-wise and normalization factors column-wise in a $m_{p \times n}$ matrix, with p genes and n cells. Z-scores (or standardization) scale all features (genes) to a mean of zero and a standard deviation of one. This approach can be utilized to facilitate, for example, the visualization of gene expression values across different genes with varying dynamic ranges of gene expression. Scaling to a zero-mean and standard deviation of one thereby is solely used to make the features (genes) comparable in a single color scale. Normalization on the other hand, in the context of scRNA-seq, aims to remove technical differences between cells such as sequencing depth or total RNA content whilst preserving true biological differences. This is a crucial step in the analysis of scRNA-seq data and we utilized the well-established 'Pearson residual' approach as implemented in the 'SCTransform' function. This approach was chosen as it has been repeatedly shown to perform well on single-cell RNAseq data (Hafemeister and Satija 2019; Lause, Berens, and Kobak 2021; Ahlmann-Eltze and Huber 2022). In addition, it should be noted that there are no conceptual differences in choosing a normalization approach for scRNA-seq or fragment-seq (formerly named sphere-seq), as at this stage of the analysis the type of data is identical. Future users of the presented method can choose a normalization approach of their liking. We updated the figure caption of the heatmap in Supplementary Fig. 3b to make the analysis approach clearer.

Mentioned in the text:

b

Supplementary Fig. 3b: Heatmap showing marker genes used for cell type annotation of liver fragment-seq experiments. The columns represent cells and the rows represent genes. Gene expression levels per cell cluster are normalized using a Pearson residual approach.

2. The sphere-seq is an unacceptable name because nothing in the dissociated blobs resembles spheres. As a diligence, I find reviewer concerns were not taken into account when responding. We would highly recommend renaming this technique to avoid future confusions.

We thank the reviewer for pointing out that there could be a misunderstanding of our method due to the name “sphere-seq”. We, therefore, now changed the name of sphere-sequencing to fragment-sequencing, which connects the two approaches of sequencing and sorting fragments of connected cells with the large fragment biosorter. We introduced the concept in the introduction of the paper.

Mentioned in the text:

Abstract: Here, we introduce fragment-sequencing (fragment-seq), a method that enables the characterization of single-cell transcriptomes within multiple spatially distinct tissue microenvironments sorted using a large fragment biosorter.

Introduction: Here, we introduce fragment-sequencing (fragment-seq), a novel method that enables the transcriptomic characterization of single cells within spatially distinct tissue niches which are partitioned using a large fragment biosorter.

3. The spatial location of where spheres came from the tissues is also very crucial. This technique is only impactful if the users can track the position of the blobs. Thus, the readers (and of course reviewers) would be interested in a proof-of-concept of the proposed approaches as described in the paper. Putting such an important direction

out-of-scope is premature from a revision perspective. This is especially needed because of direct comparisons of MERFISH, DBIT-seq etc. technologies to the proposed method. They are only fair comparisons if the spatial data is retained, measured, and incorporated into this technology.

We thank the reviewer for their comment. We would like to clarify the conceptual difference between the spatial localisation that is reconstructed by our sphere-seq (fragment-seq) approach and in slice-based spatial transcriptomics methods such as MERSCOPE and DBIT-seq.

Fragment-seq can retrieve the single-cell transcriptomes of individual 3D cellular neighborhoods (= tissue fragments), derived from random locations of the donor tissue. Each of these spatial neighborhoods can then be assigned to a particular niche (e.g. metastatic-proximal or -distal tissue, periportal or pericentral areas) - but not an exact physical location within the host tissue. Therefore, this substantially increases the power of a single-cell RNA sequencing based method to reflect niche composition, cell-cell interactions and ligand receptor interactions as they actually occur within tissues. In contrast, slice-based spatial transcriptomic methods like MERSCOPE, DBit or Visium are ideal to assign a given transcript a specific physical location within the host tissue, but have limitations that make comprehensive assessment of a niche difficult. For example, these methods have a limited throughput due to a high cost in material and time, lack 3D information (as thin tissue slices are used) and have not yet achieved single-cell whole-transcriptome resolution (either compromised by pooling cells in tiles or profiling few genes per cell). In contrast, fragment-seq provides whole transcriptome, true single-cell resolution, and incorporates 3D spatial information albeit without assigning a cell or fragment a particular physical location within the host tissue. We therefore highlight in the discussion that both types of methods complement each other, without being in direct competition.

We made additional adjustments to the text to clarify these conceptual differences.

Mentioned in the text:

Introduction:

Fragment-seq was inspired by Paired-cell sequencing (Halpern et al. 2018), PIC-seq (Giladi et al. 2020), and Clump-seq (Manco et al. 2021), which analyze spatial communities of 2-10 cells together in bulk. All three of these methods rely on computational deconvolution to approximate single-cell transcriptomes, which is inherently imprecise, especially for genes that are expressed in multiple cells. Fragment-seq refines these approaches and achieves single-cell resolution while simultaneously capturing larger communities of cells than its predecessors, thus reflecting biologically relevant tissue microenvironments in three-dimensional space. For greater analytical power, fragments can be grouped based on landmark genes or the presence/absence of one cell type to represent spatial niches. Thus, fragment-seq exploits the spatial proximity of landmark cells or genes to other cell types to more accurately reconstruct tissue niches, without assigning a specific physical location to a cell or transcript. Importantly, this enabled us to predict ligand-receptor (L-R) interactions which are not only significantly enriched in a scRNA-seq dataset, but

actually co-occur in specific microenvironments. To this end, fragment-seq combines previously established methods of single object sorting using a large fragment biosorter, cell hashing using lipid-tagged barcodes (McGinnis et al. 2019), and scRNA-seq (Macosko et al. 2015; Klein et al. 2015).

Reviewer #2 (Remarks to the Author):

We applaud the revised manuscript by the authors that much more rigorously benchmark sphere-seq based on this reviewer's feedback. There remain a few minor comments that should be addressed before this manuscript is acceptable for publication, and better inform potential users for this methodology.

Related to Q1.

The authors' response to the concern regarding the variation of the sphere size, found them "to be comparable in size and cell number (Supplementary Fig. 6a,b and Supplementary Fig. 7c,d)". The boxplot representation here should be replaced with a violin plot, as it is difficult to visually confirm that. Similarly, the term "comparable" should be toned down since there are differences observed.

We thank the reviewer for pointing out this issue and we toned down this statement. We also replaced the boxplots with violin plots. (Please note that spheres are referred to as fragments in the revised version of the manuscript.)

Mentioned in the text:

First, we compared pericentral and periportal fragments and found them to be comparable in size (Supplementary Fig. 6a), while cell counts were on average slightly lower in periportal than in pericentral fragments (Supplementary Fig. 6b). In order to test for biases introduced by fragment size or cell number, we divided the dataset according to fragment size into small (211 - 325 μm) or big fragments (326 - 457 μm) or applied different minimum cutoffs for cell numbers (≥ 5 or ≥ 20 cells per fragment). Next, we performed differential gene expression (DGE) analysis for LECs or KCs, which did not reveal any significant changes (Supplementary Fig. 6c,d). A similar pattern of significantly upregulated zoned genes could be found in all scenarios (Supplementary Fig. 6e).

Supplementary Fig. 6: Assessment of biases introduced by fragment size and cell counts per fragment. a and b, Violin plots comparing fragment sizes (a) and cell counts

per fragment (**b**) between different lobule areas (n = pericentral: 829, periportal: 714 fragments across 9 samples). P-values were calculated with a non-parametric Wilcoxon signed-rank test (NS > 0.05, * < 0.05).

Supplementary Fig. 7: Fragment-sequencing application to mouse metastatic liver. **c** and **d**, Violin plots comparing fragment sizes per sphere (**c**) and cell count per fragment (**d**) between distal and proximal areas (distal: 289; n = proximal: 51 fragments across 3 samples). P-values were calculated with a non-parametric Wilcoxon signed-rank test (NS > 0.05).

It appears from Supplementary Fig 6g that several L-R interactions were not detectable when split across the two different size groups? Perhaps this is one of the limitations of the method caused by the uncontrollable size difference. That authors should comment on this in the results and discussion.

We thank the reviewer for pointing out that some L-R interactions are not replicated across differently-sized groups. In order to assess whether this effect is caused by differential fragment sizes, or is rather the consequence of undersampling, we downsampled the data sets of CV and PV fragments from the whole size range (211-457 µm) to the same cell numbers that are within the size-restricted (211-325 µm) datasets. This demonstrated that downsampling alone also leads to the loss of some L-R interactions. For example, *Vcam1 - Integrin a9b1 complex* or *Cxcl14 - Cxcr4* are lost in both the size-restricted and downsampled data. In sum, this indicates that the overall number of acquired fragments and cells is more critical to the robustness of results than the consistency of fragment sizes. We included a new figure (Supplementary Fig. 6h) and some comments about that in the results of the manuscript.

Mentioned in the text:

Supplementary Fig. 6h: Predicted ligand-receptor (L-R) interactions between KCs and T cells in pericentral or periportal zones (n=9 samples). Interaction scores were calculated from grouped fragment-seq data using different technical cutoffs by CellPhoneDB. Interaction scores from fragments of different sizes/cell counts are highlighted by different colors of the bar borders (red: 211-325 μm, black: 211-457 μm downsampled to match the cell numbers of the 211-325 μm datasets, yellow: 211-457 μm). Interactions mentioned in the main text are highlighted with black boxes.

Methods:

L-R interaction analysis of datasets with different cutoffs was performed as described in *Ligand-receptor (L-R) interaction between different groups using CellPhoneDB*. Interaction scores of datasets with different sizes and cell number cutoffs were then plotted in barplots for comparison. To assess the influence of fragment size or sample size on L-R interaction analysis, cells from the pericentral and periportal datasets which included the whole size range (211-457 μm) were randomly downsampled to the same amounts of cells as present in the pericentral and periportal areas of the small fragment-size data sets (211-325 μm).

Results:

Moreover, recovered cell types and predicted L-R interactions were similar between different cutoffs, with only a few L-R interactions not being replicated across all size gates (Supplementary Fig. 6f,g). In order to assess whether this effect is caused by differential fragment sizes, or is rather a consequence of undersampling, we compared L-R interactions from the complete pericentral and periportal datasets with datasets from small fragment sizes (211-325 μm), and a downsampled dataset, representing the same amount of cells as the small fragment size data, but covering the complete size range (211-457 μm) (Supplementary Fig. 6h). This revealed that interactions such as *Vcam1|Itga9* and *Cxcl14|Cxcr4* were lost in both analyses (211-325 μm and 211-457 μm downsampled), indicating that the overall number of acquired fragments is more critical to the robustness of results than the consistency of fragment sizes. Therefore, we recommend prioritizing sample size over restrictive gating or subsampling of fragment sizes.

Related to Q3.

The authors did not fully address my concern related to any potential mechanical fragmentation induced transcriptional changes beyond those from scRNA-seq. The comparison in Q4 thereafter has more to do with cell type annotation and transcript counts rather than additional mechanically induced transcriptional responses due to sphere-seq. This warrants some form of attempt at showing little/no changes along with discussions.

We understand that by “mechanical fragmentation” the reviewer is referring to the tissue mincing step, after which fragments (=spheres) are sorted or picked. We would like to point out that tissue mincing is a standard first step of many single-cell dissociation protocols (Aliaghaei and Haun 2022), and is also the first step of tissue dissociation in our scRNA-seq experiments. Regardless, cells in fragment-seq may undergo somewhat different stresses, e.g. they may be exposed to fluid shear stresses when being sorted in the biosorter. To assess any potential indicators of fragment-seq-induced cellular stress, we plotted GO terms associated with cellular responses to stress. However, comparing differentially expressed genes between fragment-seq and conventional scRNAseq considering all cells or specific cell types (for example Kupffer cells) did not reveal any significantly upregulated terms (Reviewer Fig. 1).

Reviewer Fig. 1: Gene ontology terms associated with cellular stress using differentially expressed genes between fragment-seq and conventional scRNA-seq from **a**, all cells and **b**, Kupffer cells.

Related to Q4.

The authors now present Supplementary Fig 3d, which indicates NS differences of UMI and gene counts, mito/cyto ratio of genes. There seems to be a considerable difference between the methods based on the mean values (e.g. 500 vs >1000 UMI per cell; 200 vs ~300 gene counts per cell). This NS is almost certainly due to the low sample size of n=2 in scRNA-seq vs n = 9 in sphere-seq. If sample sources is really the limiting factor, one can potentially subsample or bootstrap the data to better compare their average UMI/gene counts or ratio, and better test for statistical significance.

The second question is that it seems based on these values alone, sphere-seq is better than scRNA-seq. Can the author potentially explain why it would be better, as for the

protocol cells likely go through more perturbation etc, or is it due to differences in sequencing depth etc that should be accounted for in the data presented above.

We thank the reviewer for pointing out the potentially better performance of fragment-seq. We speculate that this could be due to the increased numbers of washing steps in fragment-seq compared to conventional scRNAseq, which could remove dying cells more efficiently. The difference, however, is not due to differences in sequencing depth, because we downsampled all samples to 30k reads/cell, meaning that all cells with lower than 30k reads were removed and cells with higher than 30k reads were downsampled to 30k. We described this in the figure legend and method section. Additionally, the conventional scRNA-seq and fragment-seq experiments were not done on the same liver samples and we cannot exclude differences in overall sample quality. Therefore, while the data quality in fragment-seq appears to outperform scRNA-seq, we did not want to make such claims and instead argue that results are “comparable”, which is also shown by the comparable detection of various different cell types across assays (Figure 1f). We agree that the statistical analysis might be influenced by the lower sample size of conventional scRNA-seq, however, as our main intent was to demonstrate that the data quality obtained by fragment-seq is not significantly worse than conventional scRNA-seq, therefore, we considered additional experiments to be a disproportionate effort and cost.

Mentioned in the text:

Methods:

Conventional scRNA-seq downstream analysis

Cell count matrix generation, gene name conversion, clustering, and annotation were performed as previously described for fragment-seq downstream analysis. Annotated UMAPs were then compared between fragment-seq and conventional scRNA-seq. Quality features (median UMI counts, gene features, and ratio of mitochondrial to cytoplasmic genes) were assessed after accounting for different read depths by downsampling and only considering cells with 30000 reads. Therefore we re-run zUMIs (Parekh et al. 2018) (v2.9.4) for all samples using ‘30000’ as a counting_opts downsampling parameter.

d

Supplementary Fig. 3d, Barplots comparing median UMI counts (left), gene counts (middle), and the ratio of mitochondrial to cytoplasmic genes (right) between conventional scRNA-seq and fragment-seq (n = 2 samples for conventional scRNA-seq and 9 samples for fragment-seq). Dots represent individual samples. The upper bar limit

shows the mean across samples. The error bars show the standard deviation. P-values were calculated with a non-parametric Wilcoxon signed-rank test [> 0.05 is considered non-significant (NS)]. Reads were downsampled to 30,000 reads/cell.

Fig. 1f: UMAP visualization of mouse metastatic livers. Cells are clustered, annotated, and colored by their cell type. Cells are separated based on the protocol [conventional scRNA-seq (n = 3 samples) and fragment-seq (n = 10 samples)].

For Rev Fig2, the authors show Pearson's correlation scatter plots. I assume each dot is a gene, and the x, y axis is the average (?) normalized count for that gene in scRNA-seq and sphere-seq. Are the genes plotted here all the genes? What does this plot look like if we select variable genes in scRNA-seq + variable genes in sphere-seq, and plot them in such scatter plots? These may in part be related to my concerns in Q3 related to potential mechanically induced transcriptional changes added on by sphere-seq.

We thank the reviewer for giving us the opportunity to clarify these points. Indeed, each dot represents a gene and the x and y axis are the average normalized counts of each gene in scRNA-seq and sphere-seq respectively and all genes are plotted. Additionally, we now provide Pearson correlation plots showing the variable genes only (Rev. Fig. 3), which are also highly correlated between conventional scRNA-seq and fragment-seq. As we show in response to Q3, differentially expressed genes in fragment-seq compared to conventional scRNA-seq do not exhibit any enrichment in GO terms that are associated with cellular stress, in fact, there are no GO terms significantly enriched at all.

Rev. Fig. 2: Scatterplots showing Pearson correlation between average normalized gene expression counts of Kupffer cells (left), liver endothelial cells (middle), and macrophages/monocytes (right). All genes are considered.

Rev. Fig. 3: Scatterplots showing Pearson correlation between average normalized gene expression counts of Kupffer cells (left), liver endothelial cells (middle), and macrophages/monocytes (right). Only variable genes of conventional scRNA-seq and fragment-seq are considered.

Related to Q5.

The Supplementary fig 9e would be better serviced if it is limited to cell types that can be robustly detected in both sphere-seq and Molecular Cartography for a more meaningful comparison

We thank the reviewer for this suggestion and we removed all cell types (DCs, Cholangiocytes, Neutrophils, Kupffer/LECs, and Basophils) that did not have proper cell type markers within our Molecular Cartography panel. Additionally, we clarified this in the methods of the manuscript.

Mentioned in the manuscript:

Supplementary Fig. 9e, Zoomed-in barplot comparing the cell type proportions between Molecular Cartography (black borders) and fragment-seq (red borders) of cell types that could be robustly detected in both datasets.

Methods: *Comparison of cell type abundances between fragment-seq and Molecular Cartography.* Only cell types that could be robustly detected in Molecular Cartography, because of the presence of cell type marker genes in the gene panel, were included in this analysis. Cell-type proportions from both protocols were calculated and plotted in a barplot.

Related to Q6.

The authors raised a very good point that should be included in the discussion section.

We thank the reviewer for this suggestion and we added these points to the discussion of the manuscript.

Mentioned in the text:

Like any methodology that uses spatial reconstruction, fragment-seq is limited by prior knowledge of landmark gene expression patterns (Moor and Itzkovitz 2017) to reconstruct the position of fragments within tissues. However, this could be improved by the incorporation of additional landmark genes or reporters that indicate proximity to a certain microenvironment, for example, metastasis. Other spatial transcriptomics methods (for example laser capture microdissection or array-based approaches) could be employed to sample sections with different distances to metastases and attempt to identify landmark genes, which in turn could be used to reconstruct the distance of a sphere from the metastatic site. Alternatively, proximity-labeling systems such as sLP-mCherry, based on cell-permeable mCherry that is secreted by a sender cell to integrate into the cell membrane of neighboring cells (Ombrato et al. 2019), or cell-cell contact tracing models (Zhang et al. 2022) might have the capacity to label tumor-proximal areas with fluorescent signals that could be selectively enriched with the biosorter.

Related to Minor point Q4.

Boxplots are completely and statistically meaningless with 3 points. The upper and low quartile in there, with 3 points, do not mean anything.

As previously mentioned, the authors should replace them, even with just colored dots, or a single line indicating the mean. If the rationale is optical consistency, then one can color the dots without the need for the meaningless boxplots.

We thank the reviewer for pointing out this issue and have now changed the plots and removed the boxes.

Mentioned in the text:

Fig. 3: Fragment-seq application to investigate local differences in metastatic-proximal and -distal microenvironments. c, Dot plots representing cell type proportions of grouped fragment-seq data (n=3 samples). From left to right; macrophages/monocytes, metastatic cells, Kupffer cells (KCs), and liver endothelial cells (LECs). Dots represent individual mice and dots with black circles represent grouped fragments from proximal positions. f, Dot plots representing cell type proportions from fragment-seq showing macrophage/monocyte subtypes (n=3 samples). Dots represent individual mice and dots with black circles represent grouped fragments from proximal positions.

Related to Reviewer 3 Q2:

The authors mentioned not using individual spheres for L-R analysis but instead grouped them together. This confounds this reviewer, as it seems like this defeats the purpose of sphere-seq

We thank the reviewer for giving us an opportunity to clarify this point. We agree that performing L-R interaction analysis on individual spheres would be interesting, however, the limited number of cells recovered per sphere would likely introduce a significant amount of noise in the analyses. Instead, we opted to use the spatial proximity of cells within spheres to reconstruct niches, thus allowing us to assess a much larger group of cells for L-R interactions. This allows us to robustly identify L-R interactions that are common within that niche.

Additional comments:

The authors should discuss the following to better help readers understand the potential for the method:

1. Whether this protocol can be eventually be potentially used on freshly acquired human tissue, and some of its potential future applications.
2. For precious and limited samples, would the high cell loss rate e.g. mentioned in Reviewer3 Q1 be a limitation?
3. Are there alternative ways that can be used to group different spheres, eg a sphere with 5 cells but with 1 cancer cell is certainly different from a sphere with 50 cells but with 1 cancer cell.

1. As supplementary Fig. 8 shows, fragment-seq was applied to freshly acquired biopsies from Crohn's disease patients. Demonstrating that the fragment-seq protocol is adaptable to freshly acquired human tissues. In case the reviewer is referring to the fragment-seq approach that uses the biosorter instead of the filtering approach, we have not yet carried out such experiments but foresee no difficulties besides ensuring that experiments can be carried out in BSL2 facilities. A potential application in this regard would be to study fibrotic microenvironments in Crohn's disease patients. To this end, fragments could be clustered based on the presence/absence of activated fibroblasts to analyze their influence on other cell types. This could answer fundamental questions on the development and progression of fibrosis.
2. We agree that a high cell loss rate is a potential limitation for very small and precious samples, This limitation is somewhat mitigated by the fact that human samples tend to be comparatively larger than murine samples, however, very small samples may require pooling.
3. One alternative way to group different fragments would be to plot fragments in a UMAP rather than cells. Each fragment would be a pseudo-bulk of all the cells it contains. Similar fragments would then cluster together. This approach could be used to analyze new microenvironments that are not driven by prior knowledge of landmark genes or the presence of certain cell types.

We added some comments to the discussion.

Mentioned in the discussion:

Our preliminary data show that fragment-seq can be applied to fresh human tissues, which could be potentially used to address fundamental questions about disease mechanisms in humans. For example, fragment-seq could distinguish fibrotic and non-fibrotic microenvironments in Crohn's disease samples based on the presence or absence of activated fibroblasts or could be used to separate pro- or anti-inflammatory microenvironments within solid tumors. Of note, fragment-seq is limited by cell loss, therefore, small tissue samples (approximately $< 1\text{cm}^3$) may require pooling.

Alternatively, fragments could be grouped into different spatial microenvironments based on similarities in overall gene expression within fragments: Pseudo-bulks of fragments

could be clustered in UMAP space to investigate spatial microenvironments that are not driven by prior knowledge.

Reviewer #3 (Remarks to the Author):

The authors have satisfactorily addressed my comments/concerns through additional experiments, analyses, comparisons with published data, modifications to the text, and visual improvements to the figures. I do not have any additional major comments. I have one minor comment that I hope the authors can respond to, but I do not feel that publication of this manuscript is conditional on the response.

In reviewer figure 3, the authors report that only 50% of their single cells were kept due to Multi-seq classification. As an increase in cell number recovery would improve proportional and other analyses of Sphere-seq data, did the authors attempt the negative-cell reclassification for their data (<https://github.com/chris-mcginnis-ucsf/MULTI-seq>)? Relatedly, for the cells that were negative, was there a cell type bias relative to the positive cells? In other words, did the Multi-seq process work better for some cell types than others?

We thank the reviewer for supporting the publication of our manuscript and these suggestions. We tried using the negative-cell reclassification on our data, however, it did not improve our classification results. Comparing the cell types between negatively and positively classified cells (Rev. Fig. 4) we can indeed see that there seems to be a bias and Multi-seq might work less efficiently on immune cells, especially T and B cells, and on hepatocytes.

Reviewer Fig. 4: UMAP comparing cell type between negatively and positively classified cells.

Ahlmann-Eltze, Constantin, and Wolfgang Huber. 2022. "Comparison of Transformations for Single-Cell RNA-Seq Data." *bioRxiv*. <https://doi.org/10.1101/2021.06.24.449781>.

Aliaghaei, Marzieh, and Jered B. Haun. 2022. "Optimization of Mechanical Tissue Dissociation Using an Integrated Microfluidic Device for Improved Generation of Single Cells Following Digestion." *Frontiers in Bioengineering and Biotechnology* 10 (February): 841046.

Giladi, Amir, Merav Cohen, Chiara Medaglia, Yael Baran, Baoguo Li, Mor Zada, Pierre Bost, et al. 2020. "Dissecting Cellular Crosstalk by Sequencing Physically Interacting Cells." *Nature Biotechnology* 38 (5): 629–37.

Hafemeister, Christoph, and Rahul Satija. 2019. "Normalization and Variance Stabilization of Single-Cell RNA-Seq Data Using Regularized Negative Binomial Regression." *Genome*

Biology 20 (1): 296.

- Halpern, Keren Bahar, Rom Shenhav, Hassan Massalha, Beata Toth, Adi Egozi, Efi E. Massasa, Chiara Medgalia, et al. 2018. "Paired-Cell Sequencing Enables Spatial Gene Expression Mapping of Liver Endothelial Cells." *Nature Biotechnology* 36 (10): 962–70.
- Klein, Allon M., Linas Mazutis, Ilke Akartuna, Naren Tallapragada, Adrian Veres, Victor Li, Leonid Peshkin, David A. Weitz, and Marc W. Kirschner. 2015. "Droplet Barcoding for Single-Cell Transcriptomics Applied to Embryonic Stem Cells." *Cell*. <https://doi.org/10.1016/j.cell.2015.04.044>.
- Lause, Jan, Philipp Berens, and Dmitry Kobak. 2021. "Analytic Pearson Residuals for Normalization of Single-Cell RNA-Seq UMI Data." *Genome Biology* 22 (1): 258.
- Macosko, Evan Z., Anindita Basu, Rahul Satija, James Nemesh, Karthik Shekhar, Melissa Goldman, Itay Tirosh, et al. 2015. "Highly Parallel Genome-Wide Expression Profiling of Individual Cells Using Nanoliter Droplets." *Cell* 161 (5): 1202–14.
- Manco, Rita, Inna Averbukh, Ziv Porat, Keren Bahar Halpern, Ido Amit, and Shalev Itzkovitz. 2021. "Clump Sequencing Exposes the Spatial Expression Programs of Intestinal Secretory Cells." *Nature Communications* 12 (1): 3074.
- McGinnis, Christopher S., David M. Patterson, Juliane Winkler, Daniel N. Conrad, Marco Y. Hein, Vasudha Srivastava, Jennifer L. Hu, et al. 2019. "MULTI-Seq: Sample Multiplexing for Single-Cell RNA Sequencing Using Lipid-Tagged Indices." *Nature Methods* 16 (7): 619–26.
- Moor, Andreas E., and Shalev Itzkovitz. 2017. "Spatial Transcriptomics: Paving the Way for Tissue-Level Systems Biology." *Current Opinion in Biotechnology* 46 (August): 126–33.
- Ombrato, Luigi, Emma Nolan, Ivana Kurelac, Antranik Mavousian, Victoria Louise Bridgeman, Ivonne Heinze, Probir Chakravarty, et al. 2019. "Metastatic-Niche Labelling Reveals Parenchymal Cells with Stem Features." *Nature* 572 (7771): 603–8.
- Parekh, Swati, Christoph Ziegenhain, Beate Vieth, Wolfgang Enard, and Ines Hellmann. 2018. "zUMIs - A Fast and Flexible Pipeline to Process RNA Sequencing Data with UMIs." *GigaScience* 7 (6). <https://doi.org/10.1093/gigascience/giy059>.
- Zhang, Shaohua, Huan Zhao, Zixin Liu, Kuo Liu, Huan Zhu, Wenjuan Pu, Lingjuan He, Rong A. Wang, and Bin Zhou. 2022. "Monitoring of Cell-Cell Communication and Contact History in Mammals." *Science* 378 (6623): eabo5503.

REVIEWERS' COMMENTS

Reviewer #1 (Remarks to the Author):

The reviewer is partially satisfied with the revision.

1. The normalization approaches have been clarified.
2. Re-naming to fragment-seq is now appropriate.
3. Spatial aspect of blobs but not the exact position is understood. However, the authors propose additional tracing methods to infer spatial niches of the data. Can authors demonstrate the feasibility of one of these approaches:

However, this could be improved by the incorporation of additional landmark genes or reporters that indicate proximity to a certain microenvironment, for example, metastasis. Other spatial transcriptomics methods (for example laser capture microdissection or array-based approaches) could be employed to sample sections with different distances to metastases and attempt to identify landmark genes, which in turn could be used to reconstruct the distance of a sphere from the metastatic site. Alternatively, proximity-labeling systems such as sLP-mCherry, based on cell-permeable mCherry that is secreted by a sender cell to integrate into the cell membrane of neighboring cells⁴², or novel cell-cell contact tracing models⁴⁷ might have the capacity to label tumor-proximal areas with fluorescent signals that could be selectively enriched with the biosorter. Alternatively, fragments could be grouped into different spatial microenvironments based on similarities in overall gene expression within fragments: Pseudo-bulks of fragments could be clustered to identify spatial microenvironments that are not driven by prior knowledge.

I think for this paper such a feasibility would be great addition.

4. The GitHub should be renamed according to fragment-seq: <https://github.com/Moors-Code/Sphere-sequencing>

Reviewer #2 (Remarks to the Author):

The authors have sufficiently addressed my concerns, and recommend publication in Nature Communications. Cheers.